# The regulatory light chain mediates inactivation of myosin motors during active shortening of cardiac muscle

Thomas Kampourakis [1,2 ✉] & Malcolm Irving[1,2]

The normal function of heart muscle depends on its ability to contract more strongly at longer length. Increased venous filling stretches relaxed heart muscle cells, triggering a stronger contraction in the next beat- the Frank-Starling relation. Conversely, heart muscle cells are inactivated when they shorten during ejection, accelerating relaxation to facilitate refilling before the next beat. Although both effects are essential for the efficient function of the heart, the underlying mechanisms were unknown. Using bifunctional fluorescent probes on the regulatory light chain of the myosin motor we show that its N-terminal domain may be captured in the folded OFF state of the myosin dimer at the end of the working-stroke of the actin-attached motor, whilst its C-terminal domain joins the OFF state only after motor detachment from actin. We propose that sequential folding of myosin motors onto the filament backbone may be responsible for shortening-induced de-activation in the heart.

[1] Randall Centre for Cell and Molecular Biophysics, King's College London, London, UK. [2] British Heart Foundation Centre of Research Excellence, King's College London, London, UK. ✉email: thomas.kampourakis@kcl.ac.uk

Ejection of blood in the contraction phase of the heartbeat (systole) is tightly coupled to venous filling of the heart in the relaxed state of the heart muscle between beats (diastole). This property of the heart, usually referred to as the Frank-Starling relationship, plays a vital autoregulatory role, not least because the heart consists of two pump systems in series driving the systemic and pulmonary circulations. It is also fundamental to the response to exercise, and an abnormal Frank-Starling relationship is commonly linked with heart disease[1,2]. Since the greater volume of the heart on increased diastolic filling is geometrically related to an increase in the length of the muscle cells in the walls of its chambers, the cellular correlate of the Frank-Starling relationship is the stronger contraction of each heart muscle cell in systole when the cells have been stretched in diastole, a phenomenon referred to as length-dependent activation (LDA)[3]. The opposite effect is thought to occur when heart muscle cells shorten during the ejection phase of the cardiac cycle, accelerating the subsequent rapid relaxation of the heart muscle that allows efficient refilling of its chambers during diastole. Impairment of this effect, which we refer to as shortening-induced deactivation (SDA), is also commonly involved in heart disease[4–8]. The molecular mechanisms responsible for SDA and LDA are unknown[3,7].

At the molecular level, contraction of heart muscle is driven by the relative sliding between arrays of myosin-containing thick filaments and actin-containing thin filaments, coupled to the hydrolysis of ATP. The fundamental building block of heart muscle is the sarcomere, containing a set of laterally aligned thick filaments interdigitating with two opposed sets of thin filaments, pulling them towards the midpoint of the thick filaments during contraction. The mammalian heart works in a narrow range of sarcomere lengths, between about 1.6 and 2.3 μm, and the force produced by isolated heart muscle preparations when they contract at constant length (isometrically) increases strongly with increasing sarcomere length in this range[9].

Part of the sarcomere length-dependence of isometric force is related to structural and mechanical consequences of the arrangement of the filaments in the sarcomere[10–12], but a larger contribution comes from a greater degree of activation of the filaments at longer sarcomere length[13]. Until recently, filament activation was considered exclusively in terms of the regulatory state of the thin filaments. Each contraction is triggered by a rapid but brief increase in intracellular calcium concentration ([Ca$^{2+}$]), and some of this calcium binds to troponin in the thin filaments, activating those filaments by removing a structural constraint that blocks the binding of myosin motors to generate force[12]. The intracellular calcium transient does not change when the sarcomere length is increased between beats, at least on the short timescale relevant to the present study, but the filament system becomes more sensitive to calcium[13]. The relationship between force and calcium concentration is very steep, so a relatively small increase in calcium sensitivity produces a large increase in force. LDA is often characterised by the sarcomere length-dependence of the [Ca$^{2+}$] required for half-maximal activation in the steady-state.

It is not known how increased sarcomere length leads to increased calcium sensitivity of the contractile filaments. Elucidating that mechanism would have significance beyond the Frank-Starling relationship (LDA), the inactivation of heart muscle during ejection (SDA), and the clinical impacts of their impairment in heart disease. LDA and SDA are likely to share downstream elements with other signalling cascades that modulate contractility in the heart, including those involving phosphorylation of filament-associated proteins in response to β-adrenergic stimulation. In general, those pathways alter both the calcium sensitivity of the thin filaments and the length-dependence of that sensitivity, with additional effects on the strength and speed of contraction and relaxation[14,15]. Progress in understanding the molecular mechanisms underlying LDA and SDA may therefore have wider significance for understanding the regulation of contractility in the heart.

Many previous studies have attempted to elucidate the molecular mechanism of LDA, and several candidate mechanisms have been proposed, without any one gaining general acceptance[3,16]. Early studies focused on the calcium/thin filament regulatory pathway, but later it became clear that the thick filaments also have a regulatory role, and that the regulatory states of the thick and thin filaments are positively coupled[17,18]. In addition, recent studies suggested that, although the physiological function of LDA is produced by stretching heart muscle cells in diastole, the change in sarcomere length is not detected or coupled to the LDA signalling pathway until the filaments are subsequently activated by calcium[19,20].

In this work, we looked for the mechanisms underlying LDA and SDA in the thick filament, and more specifically in the myosin regulatory light chain (RLC), because phosphorylation of, and mutations in, the RLC have been shown to alter both thin- and thick filament-based regulation[21,22]. We focused on SDA, which is a property of calcium-activated heart muscle cells, and we imposed sarcomere shortening during calcium activation to mimic conditions during ejection in the intact heart that produce filament inactivation. The results presented below lead to a molecular structural model in which sarcomere shortening inactivates myosin motors by sequential folding of their light chain domains against the thick filament backbone, starting with the N-terminal domain of the myosin regulatory light chain.

## Results

**Structural changes in NRLC and CRLC during calcium activation of heart muscle.** The myosin regulatory light chain (RLC; Fig. 1a, b, blue) consists of two mainly alpha-helical globular lobes clasping the myosin heavy chain helix (pink) in the region of the myosin motor known as its lever arm or light-chain domain, near the junction of the two motors in each myosin molecule with its coiled-coil tail. We introduced bifunctional sulfo-rhodamine (BSR) probes into either the N- or C-terminal lobe of the RLC, subsequently referred to as NRLC and CRLC respectively, in demembranated trabeculae from rat heart by replacing about half of the native RLC in each experiment with one of the recombinant RLCs in which BSR either crosslinked helices B and C in NRLC, or was attached along the E-helix in CRLC (Fig. 1b)[23]. Bifunctional rhodamine probe attachment to RLC has no effect on the functional parameters of isolated ventricular trabeculae[20–22], and does not change cardiac myofibrillar ATPase activity in relaxing conditions (Supplementary Fig. 1), suggesting that probe attachment does not alter myosin motor function or thick filament-based mechanisms of regulation.

We measured changes in the orientation of the BSR probes in the heart muscle cells by fluorescence polarization, and expressed the results in terms of the order parameter <$P_2$>, which would be +1 if all the probes were parallel to the filament axis and −0.5 if they were all perpendicular[24]. <$P_2$> for the CRLC probe decreases on increasing the calcium concentration [Ca$^{2+}$], plotted as pCa = -log$_{10}$ [Ca$^{2+}$] in Fig. 1c, indicating that the E-helix of CRLC becomes more perpendicular to the filament axis, whereas <$P_2$> for the NRLC probe axis increases, indicating that the BC probe in NRLC becomes more parallel. These changes are consistent with release of the myosin motors from the folded helical conformation characteristic of relaxed muscle, the OFF state of the myosin filament (Fig. 1a, b, lower), on activation[25]. For simplicity the model of the OFF state in Fig. 1a has been idealised to show all the myosin motors in the same folded helical state although it is likely that only a fraction of motors are in this state in relaxing conditions[19,20,25].

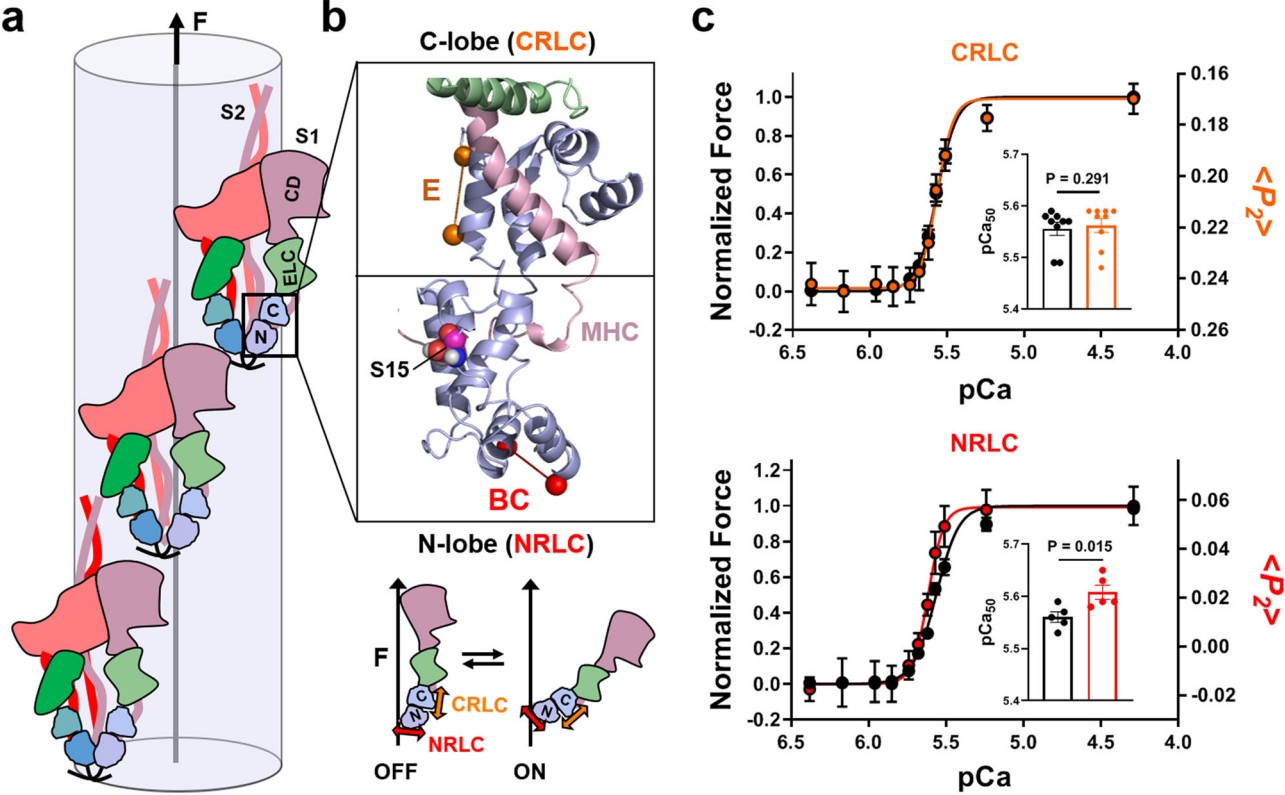

**Fig. 1 Bifunctional rhodamine probes on the myosin RLC report the regulatory state of the thick filament in cardiac muscle. a** Cartoon model of a hypothetical extreme OFF state of the cardiac thick filament in which all the myosin heads (S1) are assumed to be folded back onto their tail domains (S2) in the asymmetric interacting heads motif (IHM). The catalytic domain (CD), essential light chain (ELC) and regulatory light chain (RLC) are shown in pink, green and blue, respectively. The RLC N- (NRLC) and C-lobe (CRLC) are labelled accordingly. **b** *Top:* Atomic model of the RLC region of human cardiac myosin in the OFF state (PDB 5TBY). RLC, myosin heavy chain (MHC) and ELC are shown in blue, pink and green, respectively. The labelling positions of the RLC BC- and E-helix probe in the N- and C-lobe are indicated by red and orange spheres, respectively. The phosphorylatable serine 15 (S15) in the RLC N-terminal extension is shown in van-der-Waals representation. Bottom: Schematic demonstrating the orientation of the NRLC (red) and CRLC probe (orange) with respect to the filament axis (F) in the myosin head OFF (left) and ON states (right). Note that the NRLC and CRLC probes become more parallel and perpendicular to the filament axis, respectively, upon myosin heads leaving the OFF state. **c** Calcium-dependence of force (black) and RLC probe orientation measured as the order parameter $<P_2>$ (red and orange for NRLC- ($n = 5$ trabeculae) and CRLC-probe exchanged trabeculae ($n = 9$ trabeculae), respectively. Inset: summary of $pCa_{50}$ values (means ± s.e.m.). Statistical significance of differences between force and $<P_2>$ were assessed with a two-tailed, paired Student's $t$ test. Source data are provided as a Source Data file.

The $<P_2>$ values reported for the NRLC and CRLC probes contain contributions from all the motor conformations present in a given set of conditions. They provide a highly reproducible readout of transitions between those conformations (Fig. 1c) and the calcium dependence and kinetics of those transitions, but steady-state values of $<P_2>$ cannot in general be directly related to the fraction of motors in each conformation.

$<P_2>$ values from these BSR probes provide a highly reproducible in situ read-out of the structural changes in NRLC and CRLC on activation of the thick filament. Like force (Fig. 1c, black), these structural changes (red) are extremely sensitive to the steady level of $[Ca^{2+}]$. CRLC probe orientation has a similar $[Ca^{2+}]$-dependence as isometric force (Fig. 1c, upper; Supplementary Table 1), but NRLC orientation is more sensitive to $[Ca^{2+}]$ and has a steeper $[Ca^{2+}]$-dependence, corresponding to a Hill coefficient of about 8 (Fig. 1c, lower; Supplementary Table 1). We previously showed that the steady-state calcium dependence of NRLC probe orientation has a higher co-operativity than that of probes on troponin, and that inhibiting active force with Blebbistatin reduces the cooperativity and calcium sensitivity of the structural changes in the thin filament reported by the troponin probes[21,26]. Those results suggest that the steepness of the steady-state force-calcium relation is due to intrinsic co-operativity in thick filament activation, probably related

to intermolecular interactions between the myosin motors in the OFF state[27] (Fig. 1a), a lower degree of co-operativity in thin filament activation, and positive coupling between the activation states of the thin and thick filaments. The present results extend those conclusions showing that, unexpectedly, the steady-state calcium dependence of the structural changes in NRLC exhibit higher co-operativity than those of CRLC, suggesting that there is some flexibility between the two lobes of the RLC in this protocol and that their orientations may report different structural components of thick filament activation.

**Effect of rapid shortening and re-stretch on NRLC and CRLC orientation.** In skeletal muscle cells contracting at constant length, the imposition of rapid shortening reduces the force to zero; most myosin motors detach from actin, and both the thick filaments[28] and thin filaments[29] become partially inactivated. When a rapid re-stretch to the original length is subsequently imposed, there is a rapid force spike due to stretch of the motors that are still actin-attached at the time of the re-stretch, followed by a slower force increase associated with attachment and force generation by a new set of motors[30]. We applied this protocol to calcium-activated trabeculae to determine the associated changes in conformation of the NRLC and CRLC (Fig. 2a).

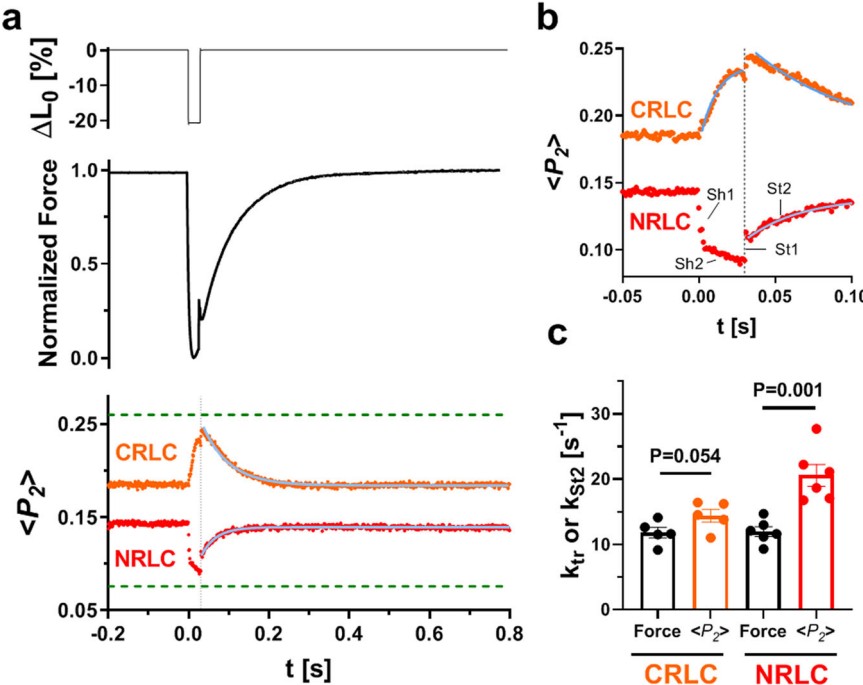

**Fig. 2 Changes in orientation of the NRLC and CRLC probes in ventricular trabeculae in response to rapid shortening and re-stretch. a** Length change ($\Delta L_O$), normalized force and $<P_2>$ in trabeculae exchanged with the NRLC (red) and CRLC probe (orange). Probe orientations in the absence of calcium (pCa 9) are indicated as green dashed lines. **b** $<P_2>$ responses on an expanded time scale. The initial fast phase (Sh1 and St1) and second slower phases (Sh2 and St2) of the NRLC probe orientation are indicated accordingly. **c** Rate constants of force re-development ($k_{tr}$) or second-phase $<P_2>$ recovery after the stretch ($k_{St2}$) (means ± s.e.m.; NRLC, $n = 6$ trabeculae; and CRLC, $n = 5$ trabeculae). Sarcomere length was set to 2.2 µm in the relaxed trabeculae. Statistical significance of differences between force and $<P_2>$ were assessed with a paired, two-tailed Student's $t$ test. Source data are provided as a Source Data file.

The orientation of the NRLC and CRLC probes changed towards their respective OFF or relaxed values (Fig. 2a, green dashed lines) during unloaded shortening. By the end of the ~30 ms shortening period (Fig. 2a, vertical dashed line), $<P_2>$ for both the NRLC and CRLC probes had recovered about two-thirds of the way towards their relaxed values ($68 \pm 5\%$ ($n = 6$) vs. $62 \pm 8\%$ ($n = 6$), respectively, mean ± s.e.m.). However, the time courses of the orientation changes were markedly different for the two probes (Fig. 2b). That of the NRLC probe had two distinct phases: a fast phase (Sh1) in the first 2–4 ms followed by a slower phase (Sh2) during the remainder of the shortening period, whereas that of the CRLC probe could be described by a single exponential with a rate constant of $108 \pm 6 \, s^{-1}$ (mean ± s.e.m., $n = 8$) (Fig. 2b). The fall of force had a rate constant of more than $250 \, s^{-1}$.

There was a similar contrast between the time courses of the responses of the two probes to a rapid re-stretch after 30 ms of unloaded shortening. The response of the NRLC probe again had two components, a step-like response complete within 1 ms (St1) followed by a slow exponential phase with a rate constant of ~$20 \, s^{-1}$ (St2; Fig. 2b), significantly faster than that for force re-development (ca $12 \, s^{-1}$; Supplementary Table 1; Fig. 2c). In contrast, the response of the CRLC probe was dominated by a single exponential recovery to the isometric level with a rate constant of ca $15 \, s^{-1}$, which is not significantly different from that of force re-development.

These experiments show that the CRLC probe primarily signals the detachment of myosin motors from actin during shortening, estimated in previous studies as ~$100 \, s^{-1}$ for cardiac myosin[31–33], and the subsequent slower ($12–15 \, s^{-1}$) re-attachment of myosin motors to actin and force development. In contrast, the change in orientation of the NRLC probe has two temporal components in the response to both shortening and re-stretch, neither of which tracks detachment or reattachment of the myosin motors per se,

suggesting that NRLC signals distinct structural transitions in the myosin motor with a potential regulatory function.

**Steady-state NRLC orientation is correlated with isometric force and sarcomere length.** Steady-state $[Ca^{2+}]$ titrations at different sarcomere lengths are often used to characterise length-dependent activation (LDA), and we have previously shown that both the NRLC and CRLC probes, like active force, exhibit higher $[Ca^{2+}]$-sensitivity in such titrations at longer sarcomere length[21,22]. Here we used a complementary approach that is more closely related to the physiological heartbeat; we imposed different extents of shortening during contraction. The initial sarcomere length was set to 2.2 µm in the relaxed trabeculae, but sarcomeres in the central region of the trabeculae (where probe orientations are measured) shorten during activation, stretching the weaker end regions[34]. We estimated sarcomere shortening in the central region from the increase in fluorescence intensity as more sarcomeres move into the measuring beam (Supplementary Fig. 2)[35]. This method indicated that the central sarcomeres shortened by $10.1 \pm 2.7\%$ (mean ± s.d., $n = 7$) on full calcium activation in the present experiments, consistent with previous direct measurements of sarcomere length by diffraction[34], so that the sarcomere length in the central region was about 2.0 µm before the imposition of shortening steps. From this starting point, active sarcomere shortening estimated by the fluorescence intensity method was proportional to the trabecular length change in the steady state (Supplementary Fig. 2c). After a total of 20% trabecular shortening imposed from an initial sarcomere length of 2.0 µm, the sarcomere length is therefore estimated to be 1.6 µm.

The steady level of force after shortening steps of between 5 and 20%, measured when trabecular length and sarcomere length (Supplementary Fig. 2) were constant, was smaller after larger

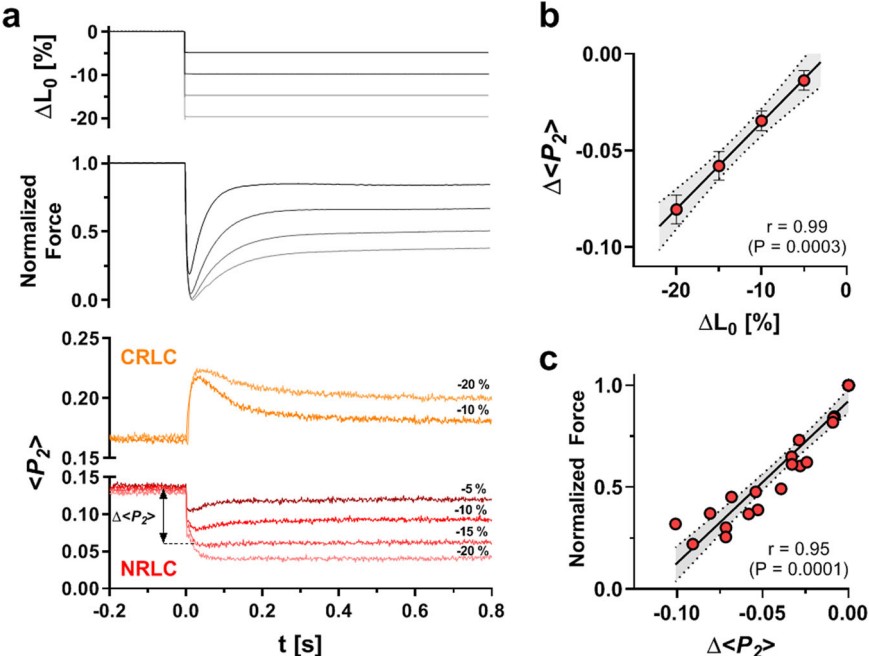

**Fig. 3 Orientation changes of the NRLC and CRLC probe in response to shortening of Ca$^{2+}$-activated muscle. a** Upper panel: muscle length change ($\Delta L_0$); middle panel: normalized force; bottom panel: $<P_2>$ from the NRLC (red) and CRLC probe (orange). **b** Length-dependence of the change in $<P_2>$ ($\Delta<P_2>$) from NRLC measured at the end of the $<P_2>$ transient. Data are shown as means ± s.e.m. ($n = 5$ for −20% $\Delta L_0$ and −10% $\Delta L_0$; and $n = 4$ for −5% $\Delta L_0$ and −15% $\Delta L_0$). **c** Plot of steady state $<P_2>$ change ($\Delta<P_2>$) of NRLC versus normalized isometric force ($n = 23$; pooled data from n = 5 trabeculae). Solid line represents linear regression with 95% confidence interval indicated by dotted lines. Pearson correlation coefficient (r) and p value for a two-tailed student's t test are shown on the bottom right. Source data are provided as a Source Data file.

shortening steps (Fig. 3a), consistent with the well-known force-sarcomere length relationship in heart muscle[5,26]. Force recovery was faster for smaller shortening steps, with fitted rate constants in the range of 6–25 s$^{-1}$. The orientations of both the NRLC (red) and CRLC (orange) probes initially changed towards their respective OFF or relaxed values after the shortening step (Fig. 3a) but with distinct time courses, as observed during the shortening phase of the shortening/re-stretch protocol. The fast phase of the NRLC response was still present for a 5% release, albeit with smaller amplitude, but the direction of the slow phase was dependent on step size. The total change in NRLC orientation at the end of the slow phase ($\Delta<P_2>$) was proportional to the size of the shortening step (Fig. 3b). Moreover, $\Delta<P_2>$ for NRLC was strongly correlated with both steady-state force (Fig. 3c) and the rate of force re-development ($k_{tr}$) at each sarcomere length (Supplementary Fig. 3), suggesting that the sarcomere-length dependence of both isometric force and the rate of force development may be primarily determined by the activation state of the myosin motors as reported by the NRLC probe.

The CRLC probe response was again distinct from that of the NRLC. The initial change in CRLC orientation after the shortening steps was well described by a single exponential with a rate constant of about 100 s$^{-1}$, as in the shortening-restretch protocol. The peak change in CRLC orientation was almost the same after 10% and 20% shortening (Fig. 3a). CRLC orientation then recovered but, like force, only partially. Both the fractional amplitude and time course of CRLC recovery matched that of force. For a 10% length step for example, the respective rate constants were 8.6 ± 2.6 s$^{-1}$ and 7.9 ± 2.2 s$^{-1}$ (mean ± s.d., $n = 3$; $P > 0.05$ for two-tailed, paired student's t test).

The regulatory structural changes in the thick filament induced by sarcomere shortening in this protocol are transmitted to the thin filament. A bifunctional rhodamine probe attached to the E-helix of troponin C (TnC), which is sensitive to binding of both

Ca$^{2+}$ and myosin motors to the thin filament[36], changed orientation towards its relaxed value at 107 ± 11 s$^{-1}$ (mean ± s.e.m., $n = 3$) after a shortening step (Supplementary Fig. 4), the same rate as measured for the CRLC probe, and corresponding to the rate of myosin motor detachment from actin. However, the partial recovery of the TnC E-helix probe signal during force re-development was faster (rate constant 16 ± 4 s$^{-1}$, mean ± s.e.m., $n = 3$) than force recovery, suggesting that the activation state of the thin filament is not simply proportional to the fraction of actin-bound myosin motors.

**NRLC orientation changes during slow ramp shortening**. The shortening steps used in the above experiments are much faster than physiological shortening during the ejection phase of the heartbeat. To discover whether the changes in NRLC and CRLC orientation seen in those experiments are present at more physiological shortening speeds, we measured changes in the probe orientations during slow ramp shortening (Fig. 4a). When 20% ramp shortening was imposed with a duration of 100, 200 or 400 ms at maximal calcium activation, force decreased in two phases, a fast phase in which both the amplitude and speed of the force change increased with shortening velocity followed by an almost linear velocity-independent phase. NRLC (red) and CRLC (orange) orientations again changed towards their relaxed values during shortening, with distinct time courses. NRLC orientation primarily tracked trabecular length, with little sensitivity to shortening velocity or force, suggesting that it depends primarily on sarcomere length rather than force during slow shortening, whereas the CRLC signal was less linear, likely reflecting its dependence on the fraction of myosin motors attached to thin filaments, as seen after length steps. The almost linear dependence of NRLC orientation on the extent of shortening was also observed, with the same slope, during contraction at lower [Ca$^{2+}$], closer to that expected for the peak of the [Ca$^{2+}$]

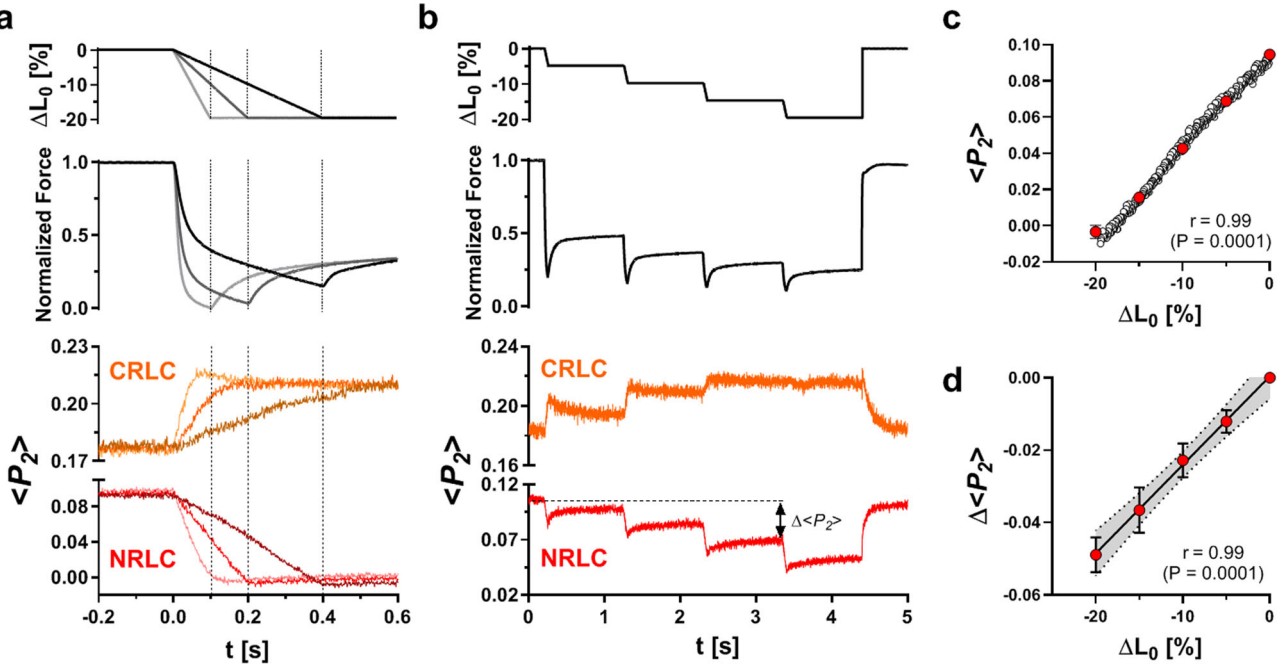

**Fig. 4 Orientation changes of the NRLC and CRLC probe in response to ramp and staircase shortening. a** NRLC (red) and CRLC probe orientation (orange) during ramp shortening of ventricular trabeculae at different velocities (2 $L_0$/s, light grey; 1 $L_0$/s, dark grey; 0.5 $L_0$/s, black). **b** Staircase ramp release protocol. Top: muscle length. Middle: force response during the mechanical protocol. Bottom: $<P_2>$ for the NRLC (red) and CRLC probe (orange). **c** Length-dependence of $<P_2>$ from the NRLC probe during ramp shortening (pooled data from **a** shown as open circles). Average $<P_2>$ at 0%, 5%, 10%, 15% and 20% shortening over all ramp shortening velocities is shown as red circles ($n = 3$ trabeculae). The Pearson correlation coefficient (r) is shown on the bottom right. Statistical significance was assessed with a two-tailed student's $t$ test. **d** Plot of $<P_2>$ change ($\Delta<P_2>$) for the NRLC probe in the staircase protocol versus muscle length change ($n = 4$ trabeculae). The Pearson correlation coefficient (r) is shown on the bottom right. Statistical significance was assessed with a two-tailed student's $t$ test. Data are shown as means ± s.e.m. Source data are provided as a Source Data file.

transient in the intact heart (Supplementary Fig. 5). These results show that NRLC orientation follows sarcomere shortening at $[Ca^{2+}]$ and shortening velocities close to those during ejection in the intact heart.

Further support for the linear length dependence of NRLC orientation was obtained using a staircase length change protocol in which a series of 5%, 50 ms ramps was applied at 1 s intervals (Fig. 4b). Each ramp produced the same change in NRLC orientation (red), whereas the change in CRLC orientation (orange), like the force change (black), became progressively smaller through the series. The total change in NRLC orientation was directly proportional to the length change in both the ramp and staircase protocols (Fig. 4c, d) and was independent of shortening velocity (and therefore of the force) in the former case (Fig. 4c, open circles).

## Discussion

The results presented above show that orientation probes on the N-terminal and C-terminal domains of the regulatory light chain of the myosin motor, NRLC and CRLC, have distinct responses to changes in sarcomere length, force and calcium concentration in heart muscle. The CLRC probe, and by inference the orientation of the C-terminal domain of the RLC, tracks the expected fraction of myosin motors attached to thin filaments in all the protocols and on all timescales studied, as expected if CRLC orientation primarily monitors the fractions of attached and detached motors with their distinct conformations. The NRLC probe, in contrast, responds to length steps with two temporal components, neither of which correlates with detachment of myosin motors from thin filaments or reattachment to them. These results suggest that the N-terminal domain of the RLC is a distinct structural and regulatory component of the myosin motor, whose conformation is more closely related to the regulatory state of the thick filament

and, intriguingly in the context of length-dependent activation, to the sarcomere length.

Perhaps the most surprising aspect of the present results is the very fast response of the NRLC probe to a shortening step or stretch, and we first consider whether this observation might be an artefact of a change in the position or orientation of the trabeculae with respect to the optical system during the length step. Such an explanation seems unlikely for several reasons. First, the $<P_2>$ orientation signals are ratiometric, and therefore insensitive to the number of probes in the illuminating beam. Second, the fast phase is specific to the NRLC probe, despite the fact that the CRLC and NRLC probes have roughly the same value of $<P_2>$, and therefore a similar mean orientation with respect to the trabecular axis, before the length step. Third, the fast phase is seen for a 5% shortening step, in which the trabecula is clearly under tension throughout the protocol (Fig. 3a). Finally, the fast phase is seen for stretches as well as releases.

It therefore seems likely that the fast phase of the NRLC probe signals a real change in NRLC orientation that is complete in about 3 ms following a shortening step. There is very little sarcomere shortening during this period; the steady velocity of shortening at low load is about 2 $L_0$/s in the present conditions[37,38] (Fig. 4), corresponding to only about 12 nm of filament sliding in 3 ms. The fast phase of the NRLC response must therefore be a pre-steady state transient associated with the decrease in force and filament stress or with the power stroke in actin-attached myosin motors. The limited time resolution of the current measurements does not allow those two processes to be separated, although it is clear that the fast phase of the NRLC signal *precedes* the detachment of the myosin motor from actin signalled by the CRLC probe. We conclude that the fast change in NRLC orientation takes place in myosin motors that are attached

to actin (or in the intramolecular partners of actin-attached motors, as discussed below).

NRLC is usually considered to be part of the lever arm of the myosin motor, which undergoes a large orientation change with respect to its actin-bound catalytic domain during the power stroke[39], suggesting that the fast phase of the NRLC signal might simply signal lever arm rotation during the power stroke. However, CRLC, which is also considered to be part of the lever arm, showed a much smaller change in orientation during the fast phase, suggesting that this interpretation is an oversimplification. Quantitative dissection of the contribution of the power stroke to the fast NRLC signal would require sub-millisecond time-resolution of the probe signals and sarcomere length, as previously achieved for single fibres from skeletal muscle[40,41]. This type of experiment has not been performed on trabeculae and would only be feasible with technical developments beyond the scope of the present study. We therefore focused on alternative explanations of the NRLC signal suggested by the distinct time courses of the NRLC and CRLC signals and the correlation of NRLC orientation with sarcomere length on slower timescales.

One such alternative explanation was suggested by our previous study of the orientations of NRLC and CRLC in trabeculae during the steady states of relaxation, isometric contraction and rigor[25]. In those experiments four bifunctional fluorescent probes were attached separately to each domain of the RLC at different locations and orientations in the local co-ordinate frame of the domain, allowing the distribution of orientations of each domain with respect to the trabecular axis to be determined. The results showed that, in heart muscle in situ, NRLC and CRLC are not part of a rigid single domain but are connected by a flexible hinge. A similar conclusion was reached in a multi-probe analysis of NRLC and CRLC orientations in skeletal muscle[42]. Flexibility between NRLC and CRLC has also been inferred from cryo-EM structures of both the actin-bound motor in the nucleotide-free or rigor complex[43], and in the folded OFF or IHM state of the myosin dimer[44–46] (Fig. 1a). Flexibility between NRLC and CRLC implies that NRLC may not be part of the lever arm in all conditions, and therefore that changes in NRLC orientation may not report the power stroke directly. Moreover, comparison of the NRLC orientation distribution deduced from the multi-probe studies in trabeculae[25] with the cryo-EM structures showed that some NRLCs stay folded against the thick filament backbone in the OFF orientation during active isometric contraction and, unexpectedly, that the OFF orientation of NRLC is similar to that in the actin-bound rigor complex.

The potential implications of those findings for interpretation of the present results can be illustrated by considering a hypothetical myosin dimer conformation in which one motor is in the actin-attached rigor or post-power stroke state determined by cryo-EM of the nucleotide free actin-motor complex[38], and the other is folded against the thick filament backbone in the 'blocked' motor conformation of the IHM determined by cryo-EM of isolated thick filaments[39,40] (Fig. 5a, upper), although the radial position of the IHM in this model should be considered as an approximation[46]. A similar myosin motor arrangement has been proposed in intact invertebrate skeletal muscle studies using X-ray diffraction[47,48]. Note that this hypothetical dimer would fit directly, without structural refinement, into the native lattice of thick and thin filaments, drawn to scale in Fig. 5a in longitudinal (left) and transverse (right) views for a thick/thin filament centre-to-centre distance of 26 nm, the almost constant value measured in a beating mouse heart throughout the cardiac cycle[49]. If the actin-attached motor of the dimer were to detach from actin and became the 'free' head of the IHM complex (Fig. 5a, lower), there would be no change in the orientation of its NRLC domain (arrowed).

The comparison of NRLC orientations in different functional states of the motor can be generalised and quantified in terms of the angles β and γ that define NRLC orientation with respect to the filament axis[25] (Fig. 5b, top), where β is the angle between the RLC D-helix and the filament axis, and γ describes rotation around the D helix. The power stroke consists of a large increase in β and a small decrease in gamma (Fig. 5b, bottom; arrow linking the pre-power stroke state 1BR1, purple diamond, to different rigour structures, squares) whereas β and γ for the cluster of published rigor structures (squares) largely overlap those for the free head in the IHM complex (orange and red circles), with the blocked head (blue and lilac circles) at slightly higher β[40].

These comparisons suggest a structural model that could explain the orientation changes in the NRLC reported here, and potentially provide a molecular explanation for SDA (Fig. 5c). In this model the power stroke in the actin-attached motor (Fig. 5c, red) brings its NRLC into a rigor-like orientation in which it can make the same atomic interactions with its partner blocked motor (green) as in the IHM. The NRLC of the post power-stroke motor could thus be captured by its partner, and consequently by the thick filament backbone, at the end of the power stroke. When the motor subsequently detaches from actin (Fig. 5c, right), this intramolecular NRLC-NRLC interaction could catalyse the sequential folding of the other domains of the motor, starting with the CRLC, to form the fully folded IHM or OFF state. Because the orientations of the free and blocked heads in the IHM are quite similar (Fig. 5b), this hypothesis should be considered as a representative of a more general class that would be consistent with the present results, in which an NRLC-NRLC dimer interaction is induced by the power stroke. For example, the NRLC of the detached motor may not already be in the blocked conformation when its actin-attached partner is in the pre-power stroke state but may be induced to enter that state by the power stroke (Supplementary Fig. 6).

This class of model has some strikingly unconventional features. NRLC is considered primarily as a filament docking and myosin dimer interaction domain, which may or may not be additionally part of the lever arm in an actin-attached motor. Some motors, and in particular some NRLC domains of some motors, remain in the folded OFF conformation during contraction, as suggested by our previous multi-probe NRLC study[22]. The functional unit of myosin is considered to be the dimer rather than the single motor, and motor-motor interactions within the dimer allow conformational changes associated with motor function per se to be coupled to myosin-based regulation. Specifically, coupling between completion of the power stroke and formation of the IHM within the dimer provides a mechanism by which active shortening could lead to motor deactivation, i.e. SDA.

Although the model was initially developed to provide a possible explanation for the fast phase of the NRLC response to a shortening step, it is consistent with the other results presented above. After the fast component, the NRLC signal continues to change in the same direction as shortening continues at low load (Fig. 2), as expected from continued accumulation of motors in the folded OFF state. When slower shortening is applied during active contraction, as would occur during the ejection phase of the heartbeat, motors would steadily accumulate in the folded OFF state at a rate roughly proportional to the shortening velocity, and therefore to an extent determined by the final length (Fig. 4a). When multiple shortening ramps of the same size are applied, a new aliquot of heads enters the folded OFF state during each ramp. When shortening ends and force redevelops at the shorter length, the NRLC signal stabilises to a steady value that is proportional to the extent of shortening and to the isometric force at that length (Fig. 3), suggesting that motors that were captured in the folded state during shortening

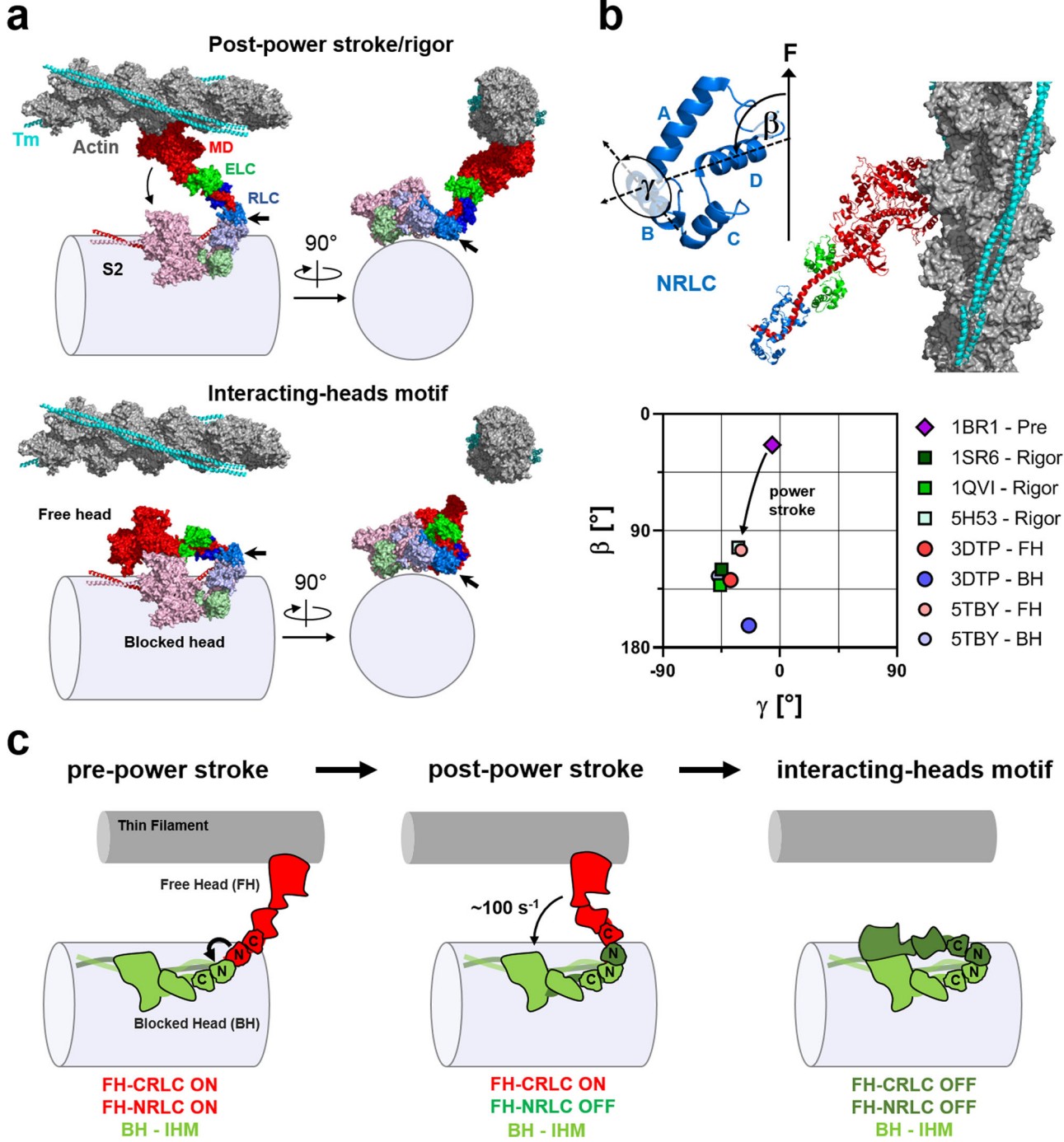

**Fig. 5 Sequential folding of cardiac myosin heads into the OFF state during active shortening. a** Top: surface representation of the free myosin head attached to actin in the rigor state and the blocked head in the IHM OFF state. Bottom: Surface representation of myosin heads folded into the interacting-heads motif (IHM). Please note that the free head can transition from the actin attached rigor state to the IHM without re-orientation of its RLC N-lobe (light blue, indicated by arrows). **b** Top: molecular reference axes used to describe the RLC N-lobe orientation. β describes the angle between the D-helix and the filament axis, and γ describes the rotation of the RLC N-lobe around the D-helix. Bottom: The (β,γ) orientation of the chicken smooth muscle myosin S1 in the pre-power stroke state (PDB 1BR1) docked onto F-actin is shown as a purple diamond. The orientation of the RLC N-lobe in different rigor myosin S1 structures (PDBs 1SR6, 1QVI, 5H53) docked onto F-actin is shown in squares. Orientations of the blocked head and free head NRLC in the IHM from tarantula skeletal (PDB 3DTP) and human cardiac muscle (PDB 5TBY) are indicated by circles. **c** Model for the sequential folding of myosin heads into the IHM OFF state during cardiac muscle active shortening. See main text for details.

remain in that state during isometric contraction at the shorter length.

When trabeculae are re-stretched rapidly during low-load shortening, the NRLC signal recovers, once again with distinct fast and slow phases (Fig. 2). Since some motors remain attached to actin during active shortening, the fast phase of the stretch response, which is smaller than that at the start of the shortening step (Fig. 2) might be explained as a reversal of the transition in Fig. 5c; the NRLC-NRLC dimer interaction in actin-attached heads near the end of the stroke could be broken by the stretch.

The slower phase of the stretch response indicates that motors that were captured in the folded OFF state during shortening can be recruited again at the longer length. Motors that are detached from actin by the stretch but do not form the folded OFF state would be expected to be available for new attachment to thin filaments. The slow phase of the NRLC response to stretch is still faster than either the orientation change in CRLC or the force (Fig. 2c), suggesting that NRLC reorientation precedes actin attachment.

However, the model in Fig. 5c has some significant limitations. It does not provide an explanation for the higher force that is generated at longer sarcomere length in a purely isometric contraction, i.e. in the absence of sarcomere shortening[9], or for the higher calcium sensitivity of the thin filaments at longer length that is responsible for the major component of the length dependence of force at the sub-maximal intracellular [Ca$^{2+}$] in intact heart muscle cells[13,50]. However, the present results shed some additional light on those effects. First, shortening-dependent inactivation of the thick filament as signalled by the NRLC probes is still present at sub-maximal [Ca$^{2+}$], with about the same amplitude as at maximal calcium activation (Supplementary Fig. 5). Second, shortening also inactivates the thin filaments, as signalled by a probe on troponin C (Supplementary Fig. 4), but with a time course similar to that of myosin motor detachment from actin, and significantly slower than the fast phase of the NRLC response to shortening. This temporal relationship suggests that either the cTnC probe is directly sensitive to the fraction of myosin motors bound to actin or the stabilisation of the folded OFF state on the thick filament following shortening triggers a slower structural change linked to inter-filament signalling. One possibility would be that the N-terminal domains of MyBP-C molecules that were bound to thin filaments at the longer length, increasing their calcium sensitivity[23], detach from actin and bind preferentially to the folded OFF state of the myosin dimer. This coupling would presumably also operate in the steady state at different [Ca$^{2+}$], and may be related to the higher calcium sensitivity of the NRLC in those conditions (Fig. 1c).

More fundamentally, positive coupling between the regulatory states of the thick and thin filaments, possibly mediated by MyBP-C as suggested above, does not in itself explain the length-dependence of the activation of either filament. The most popular explanation for LDA has been that the smaller distance between the thin and thick filaments at longer sarcomere length is the key parameter that increases thick and thin filament interaction and its sensitivity to calcium[3]. The lattice spacing hypothesis was supported by early studies which showed that osmotic compression of the filament lattice by long-chain polymers like Dextran, decreasing inter-filament separation, increases the calcium sensitivity of demembranated trabeculae to similar extent as an increase in sarcomere length that produces the same decrease in inter-filament spacing[51,52]. Some later studies cast doubt on the hypothesis by showing that the [Ca$^{2+}$] required for half-maximum activation at different Dextran concentrations was not directly correlated with inter-filament spacing measured in relaxed trabeculae at the same Dextran concentration[53]. However, the key results were not reproduced by other investigators and the hypothesis remained controversial[16,54]. Critically, calcium sensitivity and inter-filament spacing have not been measured in the same conditions. The model in Fig. 5c provides a new structural perspective on the inter-filament spacing hypothesis, because the proposed simultaneous interaction of the myosin dimer with both thick and thin filaments would be expected to be strongly sensitive to inter-filament separation, and therefore to sarcomere length.

Finally, we consider how the mechanisms discussed above might operate in the beating heart. Time-resolved X-ray diffraction experiments allow the OFF state of the thick filament to be followed during the cardiac cycle in isolated electrically-paced trabeculae contracting at constant length[19,55]. Those experiments provide direct evidence that about half of the myosin motors that are in the helical folded OFF state in diastole leave that state transiently during each heartbeat, although the other motors remain in the helically folded state, potentially providing a reserve pool of motors that are available for recruitment when an increase in contractility is required. Multiple OFF states were detected in those experiments, however, including folded states that are not helically ordered, and the results suggested that the folded helical state may be confined to the MyBP-C containing C zone of the filament in diastole, stabilised by the presence of that protein. The present NRLC and CRLC probe experiments cannot distinguish between helical and non-helical motors or identify a distinct orientation population in the C-zone, but another recent study showed that more NRLC than CRLC probes are in IHM-like conformations in diastolic conditions, suggesting that NRLC is already in that conformation in the folded non-helical state, but CRLC only joins it in the folded helical state[20], consistent with the general features of the model presented here (Fig. 5). The present results complement the X-ray studies by revealing the distinct kinetics of NRLC and CRLC during active shortening, allowing a correlation with the ejection phase of the heartbeat, in which progressive inactivation of the thick and subsequently the thin filament during shortening could explain the progressive decrease in ejection velocity[56], contributing to timely relaxation and efficient refilling. On the time scale of a single heartbeat, the capture of a myosin motor into the folded OFF state after it had executed the power stroke (Fig. 5) would exclude that particular motor from subsequent actin interaction and ATP utilisation, maximising metabolic efficiency. Increased afterload, due to arterial hypertension for example, could increase the fraction of active myosin motors through thick filament mechano-sensing[19,55], but shortening induced deactivation during ejection would still be functionally important to optimise ejection fraction, especially in the presence of diastolic dysfunction[8]. Mechano-sensing may operate in reverse during ejection or, more fundamentally, the molecular mechanism underlying mechano-sensing may be linked to the coupling between motor and regulatory states of the myosin dimer postulated above. The implications of the present results for the Frank-Starling relationship are related to LDA rather than SDA, and require further investigation of the possible links between the myosin dimer interaction model (Fig. 5), the lattice spacing hypothesis of LDA, the role of the putative binding of the N-terminus of MyBP-C to thin filaments, and thick filament mechano-sensing.

## Methods

**Protein production.** Bifunctional sulfo-rhodamine labelled RLCs were prepared as previously described[21,25,57]. Briefly, mutants of the human ventricular RLC with pairs of cysteines introduced at positions 54 and 63 crosslinking helices B and C, and 97 and 110 on helix E were obtained by site-directed mutagenesis. The mutants were expressed in BL21(DE3)-RIPL cells (Agilent Technologies) as N-terminal fusion proteins with a Histidine tag and TEV protease site from a pET6a vector. After removal of the N-terminal tag sequence by TEV protease, each of the RLC double-cysteine mutants was labelled with bifunctional sulfo-rhodamine (Invitrogen, B-10621) and purified by ion-exchange chromatography on MonoS column (GE Healthcare) to > 95% homogeneity.

**Preparation of rat right ventricular trabeculae.** All animals were treated in accordance with the guidelines approved by the UK Animal Scientific procedures Act (1986) and European Union Directive 2010/63/EU. All procedures were performed according to Schedule 1 of the UK Animal Scientific Procedure Act, 1986, which do not require ethical approval. All procedures complied with the relevant ethical regulations and were carried out in accordance with the guidelines of the Animal Welfare and Ethical Review Body (AWERB, King's College London).

Wistar rats (male, 200–250 g) were sacrificed by cervical dislocation without the use of anesthetics (Schedule 1 procedure in accordance with UK Animal Scientific Procedure Act, 1986) and demembranated right ventricular trabeculae were

prepared as described previously[36]. Briefly, hearts were removed and rinsed free of blood in Krebs solution (composition in mmol L$^{-1}$: 118 NaCl, 24.8 NaHCO$_3$, 1.18 Na$_2$HPO$_4$, 1.18 MgSO$_4$, 4.75 KCl, 2.54 CaCl$_2$, 10 glucose, bubbled with 95% O$_2$–5% CO$_2$, pH 7.4 at 20 °C). Suitable trabeculae were dissected from the right ventricle in Krebs solution containing 25 mmol L$^{-1}$ 2,3-butanedione-monoxime, permeabilized in relaxing solution (see below) containing 1% (v/v) Triton X-100 for 30 min and stored in relaxing solution containing 50% (v/v) glycerol at −20 °C for experiments.

**Fluorescence polarization experiments**. Trabeculae were mounted between a strain gauge force transducer (KRONEX, Oakland, California 94602, USA; model A-801, resonance frequency ~2 kHz) and motor (Aurora Scientific, Dublin, D6WY006, Ireland; Model 312 C). BSR-cRLCs were exchanged into demembranated trabeculae by extraction in CDTA-rigor solution (composition in mmol L$^{-1}$: 5 CDTA, 50 KCl, 40 Tris-HCl pH 8.4, 0.1% (v/v) Triton X-100) for 30 min followed by reconstitution with 40 μmol L$^{-1}$ BSR-cRLC in relaxing solution (composition in mmol L$^{-1}$: 25 Imidazole, 15 Na$_2$Creatine phosphate (Na$_2$CrP), 78.4 KPropionate (KPr), 5.65 Na$_2$ATP, 6.8 MgCl$_2$, 10 K$_2$EGTA, 1 DTT, pH 7.1) for 1 h, replacing ~50% of the endogenous cRLC[23].

Composition of experimental solutions and activation protocols were identical to those described previously for fluorescence polarization experiments[21,22]. Polarized fluorescence intensities were measured as described previously for cardiac muscle fibres[21,23,36]. Fluorescence emission from BSR-cRLCs in trabeculae were collected by a 0.25 N.A. objective using an excitation light beam in line with the emission path. The polarization of the excitation beam was switched at 1 kHz by a Pockels cell (Conoptics) between the parallel and perpendicular directions with respect to the muscle fibre long axis. The fluorescence emission was separated into parallel and perpendicular components by polarizing beam splitters, and its intensity measured by two photomultipliers, allowing determination of the order parameter $<P_2>$ that describes the dipole orientations in the trabeculae[24]. Force, muscle length and photomultiplier signals were constantly sampled at 10 kHz using dedicated programs written in LabView 2014 (National Instruments). Data were analysed using Microsoft Excel 2014 and GraphPad Prism 9.

The sarcomere length of trabeculae was adjusted to 2.2 μm by laser diffraction in relaxing solution prior to each activation. Activating solution contained (in mmol L$^{-1}$): 25 Imidazole, 15 Na$_2$CrP, 58.7 KPr, 5.65 Na$_2$ATP, 6.3 MgCl$_2$, 10 CaCl$_2$, 10 K$_2$EGTA, 1 DTT, pH 7.1. Each activation was preceded by a 2-min incubation in pre-activating solution (composition in mmol L$^{-1}$: 25 Imidazole, 15 Na$_2$CrP, 108.2 KPr, 5.65 Na$_2$ATP, 6.3 MgCl$_2$, 0.2 K$_2$EGTA, 1 DTT, pH 7.1).

Solutions with varying concentrations of free [Ca$^{2+}$] were prepared by mixing relaxing and activating solutions using MAXCHELATOR software (maxchelator.stanford.edu). Isometric force and steady-state fluorescence polarization values were measured once steady force had been established. The dependence of force and order parameters on free calcium concentration was fitted to data from individual trabeculae using non-linear least-squares regression to the modified Hill Eq. (1):

$$F = Y_0 + A \cdot \left( \left[ Ca^{2+} \right]^{n_H} / \left( -\log_{10} \left[ pCa_{50} \right]^{n_H} + \left[ Ca^{2+} \right]^{n_H} \right) \right) \tag{1}$$

where $pCa_{50}$ is the negative logarithm of [Ca$^{2+}$] corresponding to half-maximal change in $F$, $n_H$ is the Hill coefficient, $Y_0$ is the baseline, and $A$ is the amplitude (for normalized force data: $Y_0 = 0$ and $A = 1$).

Trabeculae which showed a decline in maximal calcium activated force of more than 15% after the experiments were discarded.

**Cardiac myofibrilar ATPase activity**. Cardiac myofibrils were prepared, endogenous RLCs exchanged with either wildtype RLC or RLCs crosslinked to bifunctional rhodamine, and the ATPase activity under relaxing conditions determined as previously described[22]. Briefly, cardiac myofibrils (CMFs) were prepared by homogenising freshly frozen ventricular tissue samples in myofibril buffer (composition in mmol L$^{-1}$: 20 imidazole, 75 KCl, 2 MgCl$_2$, 2 EDTA, 1 DTT, 1% (v/v) Triton X-100, pH 7.4, protease inhibitor cocktail (ROCHE), PhosStop cocktail (ROCHE)) followed by centrifugation at 5000 g for 5 min at 4 °C. CMFs were washed and homogenised three more times in the same buffer without Triton X-100. Endogenous RLCs were extracted from CMFs and reconstituted with recombinant proteins using the same protocol as described above for trabeculae. CMFs were washed three times in ATPase assay buffer (composition in mmol L$^{-1}$: 20 MOPS, 35 NaCl, 5 MgCl$_2$, 1 EGTA, 1 DTT, pH 7.0). Reactions were started by the addition of 2.5 mmol L$^{-1}$ ATP and samples were quenched with 0.5 volumes ice cold 25% (w/v) TCA solution. Samples were kept on ice at all times, diluted with double-deionized water and inorganic phosphate content measured using the malachite green assay according to manufacturer's instructions (SIGMA, GIL-LINGHAM SP8 4XT, United Kingdom; MAK030).

**Model building**. Models were built in Pymol 2.4.1 using the structures of the human β-cardiac myosin heavy meromyosin in the interacting heads motif (PDB 5TBY; https://doi.org/10.2210/pdb5TBY/pdb) and the rabbit skeletal actomyosin rigor complex (PDB 5H53; https://doi.org/10.2210/pdb5H53/pdb).

**Statistical analysis**. Two-tailed, paired student's $t$ test was used to determine the differences between force and $<P_2>$ data from the same set of trabeculae with the Prism 9 software (GraphPad Prism 9). An one-way ANOVA with Tukey's post-hoc test was used to determine the differences between three or more experimental groups (GraphPad Prism 9). A value of $p < 0.05$ was considered statistically significant. Data are represented as means ± s.d. or means ± s.e.m., with the number of independent experiments indicated by n.

**Reporting summary**. Further information on research design is available in the Nature Research Reporting Summary linked to this article.

## Data availability
The data supporting the findings of the study are available in the article and its Supplementary Information. The source data for Figs. 1, 2, 3 and 4, and Supplementary Figs. 1, 2, 3, 4 and 5 are provided as a Source data file. All remaining raw data will be available from the corresponding author upon reasonable request. Source data are provided with this paper.

## Code availability
LabView programs used for data acquisition and analysis are available from the corresponding author upon reasonable request.

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

## Acknowledgements

We are grateful to the British Heart Foundation (BHF) for financial support (FS/16/3/31887). We further thank David Trentham for help and advice.

## Author contributions

T.K. and M.I. designed research; T.K. performed research; T.K. and M.I. analyzed data; and T.K. and M.I. wrote the paper.

## Competing interests

The authors declare no competing interests.
