## [Peer Review File · Nature Communications]

REVIEWER COMMENTS

Reviewer #1 (Remarks to the Author):

In this manuscript, Kampourakis and Irving report new studies in permeabilized rat cardiac trabeculae in which the N- or C-terminal lobe of myosin regulatory light chain (RLC) have been labeled with bifunctional rhodamine. They compare the dynamic and steady state changes in these probes as trabeculae undergo activation by Ca and sarcomere length changes (primarily focusing on shortening). This experimental system reveals two very interesting differences in CRLC vs. NRLC behavior under Ca and length perturbations. First, CRLC probe orientation tracks very closely with force changes, while NRLC generally precedes force changes. The second observation is that following a sarcomere shortening event, the NRLC probe orientation reacts in two kinetic phases, both of which are faster than the force response and the CRLC probe response.

It is indeed surprising that these two regions of RLC would be behaving in distinct ways rather than in concert. The difference in kinetic rates of response is particularly significant. The authors go on to construct a very carefully reasoned conceptual model that could explain their findings. The authors conclude that length-dependent phenomena are communicated through NRLC conformation, resulting in subsequent changes in contraction force. They envision shortening as encouraging NRLC to return to its OFF conformation, thereby 'catalyzing' rapid formation of the full interacting heads motif once the myosin head detaches from the actin filament. This is a very interesting and appealing interpretation of the data, and while it does not complete the picture of length-dependent activation, it provides an important update to current paradigms. Certainly, the data presented here as well as the proposed conceptual model cannot be ignored as the mechanisms of length-dependent activation/deactivation are further elucidated.

I have three substantive comments that I feel should be addressed by the authors:

1. This report begins with steady-state behavior of the probes during activation at several different Ca concentrations. The authors emphasize an apparently higher Ca sensitivity of the NRLC probe movement relative to force, which they say demonstrates that "NRLC conformation is not tightly coupled to either force or CRLC conformation." I do not believe that this statement is entirely justified. Viewing the curves in Fig. 1C, the feet of both force and NRLC <P2> curves coincide precisely. It is only at higher Ca concentrations that the behaviors diverge. This is confirmed by the supplemental data table which reports the Hill coefficients for these data. The n_H for NRLC <P2> is significantly higher than that of the force, which in my view is an intriguing and significant result. It seems that increasing Ca activation leads to a highly cooperative and anticipatory shift of NRLC to an ON state. I suspect that the Ca effects are indirect (i.e. that increasing Ca in the presence of a myosin

inhibitor would not yield this same NRLC shift). It seems more likely that this is a manifestation of cooperativity within the thick filament, and for this reason, I think that it may be evidence of not a lack of coupling between force and NRLC but actually of a very interesting nonlinear cooperative coupling between the two. My suggestion is to consider the statement in question more carefully and possibly develop the cooperativity result a little more.

2. On a related note, it is evident from both Figure 1 and the associated supplemental table that there is a potential difference in Hill coefficient of force between the NRLC and CRLC data sets. Indeed, this seems like it could contribute to the significant difference reported in pCa_{50} of force vs. $\langle P_2 \rangle$ for NRLC. Is it possible that the NRLC probe is having an effect on thick filament structure and hence Ca-activated cooperativity?

3. In the discussion, the authors recall previous evidence from their own work suggesting that the OFF orientation of NRLC is similar to that in the actin-bound rigor complex. How does this apply to the steady-state Ca- $\langle P_2 \rangle$ curve for NRLC? Does this mean that the overall increase in $\langle P_2 \rangle$ for NRLC is a manifestation of increasing free, pre-powerstroke heads? If so, would an overall range of order parameter values be expected to be broader for CRLC, which presumably includes all free heads, pre- or post-powerstroke?

Minor points:

- Bottom of page 3, the abbreviation 'cRLC' is used – was 'RLC' intended?

- Top of page 5, the rise in force after re-stretch is described as “an exponential force increase” – I would suggest leaving out the word exponential. It's probably not worth trying to say “a force increase that follows the trajectory of a decaying exponential” or something similar.

- Figure 2C – It could be made clear in the x-axis labels that the kinetic rate in question is St_2 . The figure legend explains this, but it's nice to show this in the figure itself.

- Middle of page 14, typographical error: ...demembranated trabeculae by to similar extent...”

Stuart G. Campbell

Reviewer #2 (Remarks to the Author):

Kampourakis and Irving

Nature 2021

Review by Jim Spudich

The data presented here are a continuation of remarkable ongoing studies by Drs. Kampourakis and Irving, two leaders in muscle biology. The results presented here show that the orientation of probes on the N-terminal and C-terminal domains of the regulatory light chain of the myosin motor show distinct responses to changes in sarcomere length, force and calcium concentration in heart muscle, with the NRLC probe giving unexpected results. Using the same probes in previous studies, they showed that both the NRLC and CRLC probes, like active force, exhibit higher $[Ca^{2+}]$ -sensitivity in titrations at longer

sarcomere length (their refs 19,20). They now extend those studies.

This is a progress report of ongoing studies with probes described before, and while the data are statistically significant, beautifully presented, and unexpected, the interpretation of the results is far from clear. Thus, the statements in the abstract that the “N-terminal domain is captured in the folded OFF state of the myosin dimer” and “sequential folding of myosin motor domains onto the filament backbone is responsible for shortening-induced de-activation in the heart” are hypothetical and should not be stated as conclusions. Rather than “is”, “might be” would be more accurate.

Their results are consistent with the very interesting idea that the N-terminal domain of the RLC is a distinct structural and regulatory component of the myosin motor, whose conformation is more closely related to the regulatory state of the thick filament and, intriguingly in the context of length-dependent activation, to the sarcomere length. Given the concerns below, however, more work is warranted to be sure their NRLC probe is not artifactually affecting the system, and the paper needs to be much clearer about what aspects of their model are well established and where questions remain.

Primary issues:

1. The BSR probe for the CRLC is along a single helix and that probe behaves as one would expect during cycling. The NRLC probe, however, crosslinks helices B and C in NRLC, and this probe is reporting unexpected, potentially interesting, but possibly artifactual results. The authors must have been concerned that crosslinking two helices in the NRLC may alter function where those two helices may normally have to breathe a little during a contraction cycle. A test of this would be to see if one gets the same results with NRLC probes that bind just along a single helix. Did I miss these controls in the paper?

2. A related question to #1 above is whether the authors have any data as to whether their various probes perturb the function of purified cardiac myosin, especially with regard to the putative folding back of the heads into an OFF SRX state. This seems like it would be a good complementary study to their elegant measurements with mouse trabeculae. We, for example, have two different in vitro assays testing for the effects of HCM mutations on the putative folded back state, and could easily examine the effects of their probes on purified wild type human beta-cardiac myosin in this respect. No perturbations in such studies would be good for the authors to be able to report – if the authors are interested, they should feel free to contact me.

3. The authors speak to the folded back state as if it is a foregone conclusion that their model in Figure 1 is well established. Their model presupposes that myosin heads are flipping back and forth between extended actin-interacting heads and IHM folded back heads in each beat of the heart. There are two sources of data that do not support this model. One is Roger Cooke's original data defining SRX by mantATP experiments that show two distinct populations of myosin heads in relaxed cardiac fibers, about 50% only having the SRX signature of very low basal ATPase activity. Assuming Cooke's SRX state reflects a folded back off state (presumed to be the IHM state, but not proven), his data indicates that not all of the heads are in the IHM OFF state at rest. Furthermore, upon initiation of contraction in cardiac fibers, the % heads in the SRX state does not change. This is consistent with the idea that many heads (possibly ~50%) are in a folded back OFF state, being held out of play, and not being used for contraction, unless needed by an increased demand on the muscle, in which case possibly RLC phosphorylation would release heads from this storage pool. A second source of information comes from our purified protein experiments where we see ~50% of heads in an SRX state, consistent with Cooke's studies with fibers, and the SRX heads are not in rapid equilibrium with open ON state heads. Thus, our studies are also consistent with the OFF state being a pool of heads that are sequestered and out of play during contraction until needed by a higher demand on the muscle. Can the authors please describe their view on these two different ways of viewing the SRX OFF state. Minimally they need to make it more clear that Fig 1 is a model and not a confirmed reality, as they seem to imply.

Other issues:

Fig 1c: normalized F. What are the actual force values?

Fig 5: My understanding about the spacing between thick and thin filaments in human cardiac fibers is that there is a 20-nm gap between the heads on the thick filament while the heads are only ~15 nm long. Thus, S2 must leave the thick filament backbone to allow the S1 head to bind to actin in human fibers. If their drawing is the correct spacing in mouse fibers, their model for mice may be consistent, but their model would not fit the human fiber spacings. The authors should speak to this point.

If this is a difference between mouse and human, what would they expect to see if they repeated these experiments with human trabeculae, which would not allow for the Fig 5 interpretation?

Jim Spudich

Reviewer #3 (Remarks to the Author):

The underlying mechanisms of the healthy heart muscle which contracts strongly to eject blood during contraction (systole) and relax between beats (diastole) while refilling with blood are unknown. Contraction during systole and relaxation during diastole are associated to the shortening and stretching of the heart muscle.

Kampourakis and Irving are leading scientists in this field with a long-standing trajectory. In their manuscript, they propose an structural explanation for the shortening-induced deactivation (SDA) of the heart during diastole which sequentially involves the mediation of the regulatory light chains (RLCs) of the two heads of the myosin molecule, organized in the relaxed state as the blocked and free heads of the myosin interacting-heads motif (IHM).

This proposal is a significant contribution to the literature of this field as they propose a model based on structural information to explain the SDA, focused on the sequential folding of the myosin heads onto the thick filament backbone, being mediated specifically by the regulatory light chains (RLCs) of both myosin heads. This proposal should be of interest to a wider field of readers as it involves the well-known Frank-Starling relationship for the heart, so it should certainly influence the thinking on the field. To support their proposed model, the authors did a comprehensive study of the orientation of bifunctional fluorescent probes, inserted in the RLCs of the IHMs of rat demembrated ventricular trabeculae, during Ca²⁺-induced contraction; together with rapid shortening and re-stretching to mimic SDA.

---It would improve the manuscript if the major points detailed below are addressed.

Major points:

(1) In Fig. 1a the hand of the helix of IHMs shown is reversed. With the IHMs asymmetric conformation as shown in the figure, the sub-fragment 2 (S2) of the myosin tail of the IHM point towards the bare zone at the top, so the helix should run to the right, not to the left.

---This figure (Fig. 1a) should be corrected.

Note: As lines are not numbered, the comments below refer to the actual page quoting the specific text.

(2) On the last paragraph of page 3 it is stated that: "We introduced bifunctional sulfo-rhodamine (BSR) probes into either the N- or C-terminal lobe of the RLC, subsequently referred to as NRLC and CRLC respectively, in demembrated trabeculae from rat heart by replacing about half of the native RLC with recombinant cRLCs in which BSR either crosslinked helices B and C in NRLC, or was attached along the E-helix in CRLC (Fig. 1b)".

---The authors should address a possible disruption of the IHM due to the insertion of these probes. Does this possibility have been ruled out and if so, how? Also, it should be clarified if only one probe or the two probes were inserted on one or both heads of the IHM. As half of the RLCs are replaced, do they know whether these go on randomly, or could they possibly all go onto the free heads, and how would this affect the interpretation?

(3) For assessing the angles of the probes, the authors measured the fluorescence polarization of the heads in the demembrated muscle by the order parameter $\langle P2 \rangle$. This value should be the average value of the contribution of all the labeled heads present in the measuring beam as shown in the Suppl. Fig. 1a.

a.- In the case of the relaxed state (Fig 1c, pCa low) as it is a demembrated muscle in the relaxed state the IHM should be present, as have been shown either in isolated relaxed thick filaments of mouse and human cardiac muscle (Zoghbi et al. PNAS 2008) (AL-Khayat et al. PNAS 2013) or in demembrated and live relaxed tarantula skeletal muscle (Padron et al PNAS 2020). Therefore, the measured $\langle P2 \rangle$ should be the average of the $\langle P2 \rangle$ values for each head of the IHM reported by the CRLC and NRLC probes as present in: (1) the blocked head and (2) the free head (when docked on its partner blocked head) completing the IHM. As both heads are roughly parallel to the filament axis then the average $\langle P2 \rangle$ for CRLC should be roughly parallel and the one for NRLC roughly perpendicular. So, in the relaxed conditions both probes independently reflect the angles roughly present on the IHM, making the analysis of the probes angles straightforward, with the proviso of discarding the transient changes when the free heads sway away and back (Brito et al. J Mol Biol 2011). However, at higher pCa values when force start to increase and when it reach the plateau, it have been found that in tarantula skeletal muscle during a tetanus the blocked heads remain docked on its S2 near the backbone as it were in the relaxed state, while the free heads are released and bound to the thin filaments (Padron et al. PNAS 2020). If something similar occurs in the demembrated rat trabeculae when producing force, as the authors propose in their model (Fig. 5), this means that for the higher pCa in Fig. 1c, the measured $\langle P2 \rangle$ values could be an average of: (1) the blocked head docked on its S2 reporting a parallel angle from CRLC and a perpendicular one from the NRLC, together with (2) the opposite values from a released free head with a perpendicular angle from CRLC and parallel one from NRLC. So, the measured $\langle P2 \rangle$ average value will report mixed contradictory parallel and perpendicular angles from both types of heads.

---The authors should consider and discuss how the previous considerations affect their interpretation of the measured $\langle P2 \rangle$ values during force production at high pCa (Fig. 1) as well as on the rapid shortening and re-stretching (Fig. 2, 3, 4).

b.- In the three crowns that form the vertebrate cardiac thick filament repeat there are two crowns with three IHMs present on each one, while in the remaining crown the presence of the IHM is weaker or possibly the six heads it contains are disordered (Zoghbi et al. PNAS 2008) (AL-Khayat et al. PNAS 2013). Does the average $\langle P2 \rangle$ coming from probes within the two crowns with IHMs and from the third disordered crown report mixed contradictory parallel and perpendicular angles?

---The authors should consider and discuss how the previous considerations affects their interpretation of the measured $\langle P2 \rangle$.

c.- Accessory proteins like cMyBP-C are present on the C-Zone of the thick filaments but not on the P- and D-zones. Does the average $\langle P2 \rangle$ coming from probes within these three zones report mixed contradictory parallel and perpendicular angles?

--The authors should consider and discuss how the previous considerations affects their interpretation of the measured $\langle P2 \rangle$.

(4) In the Discussion there are several paragraphs in which knowing how labeled probes are distributed between the blocked and free heads is important as well as how relevant is this distribution on the measured $\langle P2 \rangle$ changes, as these changes are averages of reports from probes mixed between both types of heads:

1.- On page 10, first paragraph it is stated that “The CLRC probe, and by inference the orientation of the C-terminal domain of the RLC, tracks the expected fraction of myosin motors attached to thin filaments in all the protocols and on all timescales studied, as expected if CRLC orientation primarily monitors the fractions of attached and detached motors with their distinct conformations.”

-- The authors should consider and discuss how their interpretation of the measured $\langle P2 \rangle$ is affected when considering that a fraction of heads could remain bound to the backbone.

2.- On page 11, second paragraph it is stated that “Flexibility between NRLC and CRLC has also been inferred from cryo-EM structures of both the actin-bound motor in the nucleotide-free or rigor complex⁴¹, and in the folded OFF or IHM state of the myosin dimer^{42,43} (Fig. 1a).”

-- The authors should consider quoting here the papers that addressed the RLC-RLC flexibility which are Brito et al. J. Mol. Biol. 2011 and Alamo et al. J Mol. Biol. 2016.

3.- On the second paragraph in page 11 it is stated that “Moreover, comparison of the NRLC orientation distribution deduced from the multi-probe studies in trabeculae²⁵ with the cryo-EM structures showed that some NRLCs stay folded against the thick filament backbone in the OFF orientation during active isometric contraction and, unexpectedly, that the OFF orientation of NRLC is similar to that in the actin-bound rigor complex.”, and on the third paragraph “The potential implications of those findings for interpretation of the present results can be illustrated by considering a hypothetical myosin dimer conformation in which one motor is in the actin-attached

rigor or post-power stroke state determined by cryo-EM of the nucleotide free actin-motor complex⁴¹ and the other is folded against the thick filament backbone in the 'blocked' motor conformation of the IHM determined by cryo-EM of isolated thick filaments^{43,44} (Fig. 5a, upper)." In fact, the presence of the blocked head docked on its S2 and backbone during contraction of tarantula skeletal muscle have been suggested by time-resolved low-angle X-ray diffraction studies (Padron et al. PNAS 2020), as well as in the relaxed state when inducing disordering of the free heads by low temperature (Ma et al J. Gen. Physiol. 2021).

--- The authors should consider quoting the papers that addressed what they stated in the two mentioned paragraphs.

4- In Fig. 5 (cf. Suppl. Fig. 5) the authors propose a model based on the sequential folding of cardiac myosin heads into the OFF state during active shortening based on the measured average $\langle P2 \rangle$ from all heads present in the measuring beam. While the evidence from the probes in the last step (interacting-heads motif) are unambiguous (see point 3a above), the evidence for the first two steps (pre-power stroke and post-power stroke) could have a mixed contradictory origin.

--- The authors should assess the implications that the mentioned mixed contradictory origin has in the first two steps of their model.

Minor points:

1.- In Fig 2 correct uppercase in (C), it should be (c).

Raul Padron

GENERAL COMMENTS

We would like to thank both the Editors and Reviewers for their constructive criticism of our manuscript, and we have replied to the specific comments raised by the reviewers on a point-by-point basis below.

REVIEWER COMMENTS

Reviewer #1 (Remarks to the Author):

In this manuscript, Kampourakis and Irving report new studies in permeabilized rat cardiac trabeculae in which the N- or C-terminal lobe of myosin regulatory light chain (RLC) have been labeled with bifunctional rhodamine. They compare the dynamic and steady state changes in these probes as trabeculae undergo activation by Ca and sarcomere length changes (primarily focusing on shortening). This experimental system reveals two very interesting differences in CRLC vs. NRLC behavior under Ca and length perturbations. First, CRLC probe orientation tracks very closely with force changes, while NRLC generally precedes force changes. The second observation is that following a sarcomere shortening event, the NRLC probe orientation reacts in two kinetic phases, both of which are faster than the force response and the CRLC probe response.

It is indeed surprising that these two regions of RLC would be behaving in distinct ways rather than in concert. The difference in kinetic rates of response is particularly significant. The authors go on to construct a very carefully reasoned conceptual model that could explain their findings. The authors conclude that length-dependent phenomena are communicated through NRLC conformation, resulting in subsequent changes in contraction force. They envision shortening as encouraging NRLC to return to its OFF conformation, thereby 'catalyzing' rapid formation of the full interacting heads motif once the myosin head detaches from the actin filament. This is a very interesting and appealing interpretation of the data, and while it does not complete the picture of length-dependent activation, it provides an important update to current paradigms. Certainly, the data presented here as well as the proposed conceptual model cannot be ignored as the mechanisms of length-dependent activation/deactivation are further elucidated.

I have three substantive comments that I feel should be addressed by the authors:

1. This report begins with steady-state behavior of the probes during activation at several different Ca concentrations. The authors emphasize an apparently higher Ca sensitivity of the NRLC probe movement relative to force, which they say demonstrates that "NRLC conformation is not tightly coupled to either force or CRLC conformation." I do not believe that this statement is entirely justified. Viewing the curves in Fig. 1C, the feet of both force and NRLC curves coincide precisely. It is only at higher Ca concentrations that the behaviors diverge. This is confirmed by the supplemental data table which reports the Hill coefficients for these data. The n_H for NRLC is significantly higher than that of the force, which in my view is an intriguing and significant result. It seems that increasing Ca activation leads to a highly cooperative and anticipatory shift of NRLC to an ON state. I suspect that the Ca effects are indirect (i.e. that increasing Ca in the presence of a myosin inhibitor would not yield this same NRLC shift). It seems more likely that this is a manifestation of cooperativity within the thick filament, and for this reason, I think that it may be evidence of not a lack of coupling between force and NRLC but actually of a very interesting nonlinear cooperative coupling between the two. My suggestion is to consider the statement in

question more carefully and possibly develop the cooperativity result a little more.

The main focus of this paper is the dynamic change in RLC lobe orientation in response to shortening rather than its steady state calcium dependence, which was more the focus of our previous papers which also compared the calcium dependence of an RLC probe with that of probes on troponin in the thin filaments, including the effects of a myosin inhibitor, Blebbistatin, and RLC phosphorylation (Kampourakis et al., 2016, PNAS; Zhang et al., 2017, eLife). Those papers already showed that the calcium dependence of the NRLC probe has a higher cooperativity than that of probes on troponin, and that myosin inhibitors reduce the cooperativity and calcium sensitivity of structural changes in the thin filament. Those results suggest that the force-calcium relation is the consequence of thin and thick filaments acting as a coupled system, and that there is a higher degree of co-operativity in the thick than the thin filament (as also suggested here by the reviewer). We did not include these points in the submitted version of the manuscript, but we agree with the reviewer that they are directly relevant to the interpretation of the new results, and we have now expanded the relevant sections of the Results and Discussion to clarify these points, and included a reference to Zhang et al., 2017, Elife.

2. On a related note, it is evident from both Figure 1 and the associated supplemental table that there is a potential difference in Hill coefficient of force between the NRLC and CRLC data sets. Indeed, this seems like it could contribute to the significant difference reported in pCa50 of force vs. for NRLC. Is it possible that the NRLC probe is having an effect on thick filament structure and hence Ca-activated cooperativity?

The difference in Hill coefficient n_H for force between the NRLC and CRLC groups of trabeculae is not statistically significant at the $p=0.05$ level ($P=0.23$ for a two-tailed, unpaired student's t -test). Moreover, the Hill coefficients of force for both the NRLC (n_H of ~ 5.6) and CRLC probe (n_H of ~ 6.9) are not significantly different from that for force in native demembranated rat ventricular trabeculae (n_H of ~ 6 ; Kampourakis et al., 2016, PNAS). This also partially overlaps with point #2 of reviewer #2.

3. In the discussion, the authors recall previous evidence from their own work suggesting that the OFF orientation of NRLC is similar to that in the actin-bound rigor complex. How does this apply to the steady-state Ca- curve for NRLC? Does this mean that the overall increase in for NRLC is a manifestation of increasing free, pre-powerstroke heads? If so, would an overall range of order parameter values be expected to be broader for CRLC, which presumably includes all free heads, pre- or post-powerstroke?

The orientation distributions for both the N- and C-terminal lobe of RLC as determined by fluorescence polarization from multiple probes on RLC and maximal entropy formalism identify multiple populations (three for N-lobe and four for the C-lobe) with distinct orientations in relaxing conditions, isometric contraction and rigor (Kampourakis et al., 2015, Biophys J). Some of these populations match the orientations of the N-terminal and C-terminal lobe in atomic models of the myosin interacting heads motif. The other populations are likely to be associated with other biochemical and structural states of the myosin heads, and perhaps with distinct myosin component orientations in different regions of the thick filament (e.g. C-zone vs non C-zone). The measured $\langle P_2 \rangle$ values for the NRLC and CRLC probes have contributions from all these populations; the calcium titrations shown in Figure 1c signal a

change in the distribution between the different molecular conformations but cannot be identified with a single population. It seems very likely that NRLCs leave the OFF conformation when $[Ca^{2+}]$ is increased, and any NRLCs that transition from the IHM to the post-powerstroke conformation would not contribute to the observed change in $\langle P_2 \rangle$. However, because of the presence of myosin populations in other conformations, we cannot infer that the heads that leave the IHM must move to a pre-powerstroke conformation. We have edited the results chapter to further emphasize these points.

Minor points:

- Bottom of page 3, the abbreviation 'cRLC' is used – was 'RLC' intended?

Yes. Thank you.

- Top of page 5, the rise in force after re-stretch is described as “an exponential force increase” – I would suggest leaving out the word exponential. It's probably not worth trying to say “a force increase that follows the trajectory of a decaying exponential” or something similar.

Done. Thank you.

- Figure 2C – It could be made clear in the x-axis labels that the kinetic rate in question is St2. The figure legend explains this, but it's nice to show this in the figure itself.

Done. Thank you.

- Middle of page 14, typographical error: ...demembranated trabeculae by to similar extent...”

Done. Thank you.

Stuart G. Campbell

Reviewer #2 (Remarks to the Author):

Kampourakis and Irving

Nature 2021

Review by Jim Spudich

The data presented here are a continuation of remarkable ongoing studies by Drs. Kampourakis and Irving, two leaders in muscle biology. The results presented here show that the orientation of probes on the N-terminal and C-terminal domains of the regulatory light chain of the myosin motor show distinct responses to changes in sarcomere length, force and calcium concentration in heart muscle, with the NRLC probe giving unexpected results. Using the same probes in previous studies, they showed that both the NRLC and CRLC probes, like active force, exhibit higher $[Ca^{2+}]$ -sensitivity in titrations at longer sarcomere length (their refs 19,20). They now extend those studies.

This is a progress report of ongoing studies with probes described before, and while the data are statistically significant, beautifully presented, and unexpected, the interpretation of the results is far from clear. Thus, the statements in the abstract that the “N-terminal domain is captured in the folded OFF state of the myosin dimer” and “sequential folding of myosin motor domains onto the filament backbone is responsible for shortening-induced de-activation in the heart” are hypothetical and should not be stated as conclusions. Rather than “is”, “might be” would be more accurate. Their results are consistent with the very interesting idea that the N-terminal domain of the RLC is a distinct structural and regulatory component of the myosin motor, whose conformation is more closely related to the regulatory state of the thick filament and, intriguingly in the context of length-dependent activation, to the sarcomere length. Given the concerns below, however, more work is warranted to be sure their NRLC probe is not artifactually affecting the system, and the paper needs to be much clearer about what aspects of their model are well established and where questions remain.

We edited the Abstract as suggested.

Primary issues:

1. The BSR probe for the CRLC is along a single helix and that probe behaves as one would expect during cycling. The NRLC probe, however, crosslinks helices B and C in NRLC, and this probe is reporting unexpected, potentially interesting, but possibly artifactual results. The authors must have been concerned that crosslinking two helices in the NRLC may alter function where those two helices may normally have to breathe a little during a contraction cycle. A test of this would be to see if one gets the same results with NRLC probes that bind just along a single helix. Did I miss these controls in the paper?

We previously calculated the orientation of eight different bifunctional RLC probes in six different crystal structures of either isolated myosin S1 in different nucleotide states or light chain domains. We found that that the orientations of the B- and C-helices is almost identical in the six structures (average angular coordinates (θ, ϕ) for the BC-helix probe in the RLC DB-reference frame were $80.0 \pm 10.4^\circ$ and $124.8 \pm 5.0^\circ$, mean S.D., $n=6$; Kampourakis et al., 2015, Biophys J, Fig. S3). This similarity suggests that the relative orientation of the B- and C-helices, and that of the BC probe with respect to the protein frame are tightly constrained by the tertiary fold of the N-lobe. Moreover, the bifunctional rhodamine probes are attached to

the RLC via flexible linkers, and the attachment points were chosen with optimal separation minimise possible effects of probe attachment on protein conformation.

The conclusion that neither the BC nor the E-helix BSR probe (relating to the reviewer's next point) alters functionally important aspects of NRLC structure is supported by several published observations.

- 1. The calcium sensitivity of force development (pCa_{50} of ~ 5.6), Hill coefficient (n_H of ~ 6), maximal calcium-activated isometric force ($30\text{-}35\text{ mN mm}^{-2}$) and the rate of force redevelopment (k_{tr} of $\sim 12\text{ s}^{-1}$) are all the same in NRLC- and CRLC-exchanged ventricular trabeculae (Fig. 1), and in control experiments with native trabeculae (please see Kampourakis et al., 2016, PNAS; Kampourakis et al., 2018, JMCC).*
- 2. Both probes preserve the length-dependent increase in calcium sensitivity and maximal isometric force of demembrated ventricular trabeculae (Kampourakis et al., 2016, PNAS; Kampourakis et al., 2018, JMCC), which is associated with greater activation of the thick filament at longer sarcomere length (Zhang et al., 2017, Elife). A perturbed length-dependent activation response would be expected if probes altered the regulatory state of the thick filament, as shown for example for a non-truncating missense variant in RLC (R58Q; please see Kampourakis et al., 2018, JMCC), which abolishes the increase in maximal isometric tension upon sarcomere stretch.*
- 3. The temperature dependence of thick filament structure in relaxed, demembrated trabeculae is preserved after the introduction of the NRLC and CRLC probes (Park-Holohan et al., 2021, PNAS). Both probes signal a higher fraction of myosin motors in the OFF state at higher temperature with almost identical transition temperatures, in good agreement with published X-ray diffraction results from demembrated cardiac muscle cells (Xu et al 2006 Biophys J) and mammalian skeletal muscle fibres (Caremani et al., 2021, JGP).*

We expanded the first paragraph of Results to reference the key previous studies and clarify these points.

2. A related question to #1 above is whether the authors have any data as to whether their various probes perturb the function of purified cardiac myosin, especially with regard to the putative folding back of the heads into an OFF SRX state. This seems like it would be a good complementary study to their elegant measurements with mouse trabeculae. We, for example, have two different in vitro assays testing for the effects of HCM mutations on the putative folded back state, and could easily examine the effects of their probes on purified wild type human beta-cardiac myosin in this respect. No perturbations in such studies would be good for the authors to be able to report – if the authors are interested, they should feel free to contact me.

We used a simpler assay that we already had in hand to address this point, and added the new Results as Supplementary Fig. 1. They show that resting ATPase activity of cardiac myofibrils was not altered by exchange of either wildtype RLC, or RLC labelled with either the NRLC or CRLC probe. We conclude that the probes do not perturb the thick filament OFF state and by implication the interacting-heads motif.

3. The authors speak to the folded back state as if it is a foregone conclusion that their model in Figure 1 is well established. Their model presupposes that myosin heads are flipping back and forth between extended actin-interacting heads and IHM folded back heads in each beat of the heart. There are two sources of data that do not support this model. One is Roger Cooke's original data

defining SRX by mantATP experiments that show two distinct populations of myosin heads in relaxed cardiac fibers, about 50% only having the SRX signature of very low basal ATPase activity. Assuming Cooke's SRX state reflects a folded back off state (presumed to be the IHM state, but not proven), his data indicates that not all of the heads are in the IHM OFF state at rest. Furthermore, upon initiation of contraction in cardiac fibers, the % heads in the SRX state does not change. This is consistent with the idea that many heads (possibly ~50%) are in a folded back OFF state, being held out of play, and not being used for contraction, unless needed by an increased demand on the muscle, in which case possibly RLC phosphorylation would release heads from this storage pool. A second source of information comes from our purified protein experiments where we see ~50% of heads in an SRX state, consistent with Cooke's studies with fibers, and the SRX heads are not in rapid equilibrium with open ON state heads. Thus, our studies are also consistent with the OFF state being a pool of heads that are sequestered and out of play during contraction until needed by a higher demand on the muscle. Can the authors please describe their view on these two different ways of viewing the SRX OFF state. Minimally they need to make it more clear that Fig 1 is a model and not a confirmed reality, as they seem to imply.

We agree with the reviewer that not all myosin heads are likely to be in the IHM in heart muscle in relaxing conditions. This was an unintended implication of the simplified model in Fig. 1a in the submitted paper, and we have now made explicit that the model is a hypothetical fully OFF state and is not intended to represent the distribution of myosin head conformations in relaxing conditions. The existence of multiple populations of RLCs with different orientations in those conditions was established by our previous experiments with multiple probes on each lobe of the RLC (Kampourakis et al., 2015, Biophys J). Moreover, we recently showed that, in near physiological diastolic conditions, the fraction of N-lobes in an IHM-like conformation is likely to be greater than the fraction of C-lobes in that state (Park-Holohan et al., 2021, PNAS). We would also distinguish between 'folded helical' and 'folded non-helical' OFF states (Brunello et al., 2020, PNAS), which can be distinguished and quantified separately by X-ray diffraction in electrically paced trabeculae. In our view those experiments provide compelling structural evidence that there is a population of myosin heads in heart muscle cells that leave the OFF states during systole and rejoin them in diastole. Because that population is likely to be small (Brunello et al., 2020, PNAS estimate that the fraction of motors attached to actin at the peak of systole is only 10% of the total), this conclusion is not necessarily inconsistent with Cooke's concept of a pool of OFF heads in activating conditions that are available for recruitment, by RLC phosphorylation for example. This distinction, between dynamic transitions to/from the IHM state on the timescale of the heartbeat and slower changes in response for example to neurohumoral control of the heart has now been made more explicit in the final paragraph of the Discussion.

As noted in the reply to point 3 of reviewer 1, the $\langle P_2 \rangle$ order parameters are sensitive to changes in the distribution between all the different RLC conformations present in a particular set of conditions or protocol but cannot be identified with a single population. Therefore, the changes in $\langle P_2 \rangle$ reported here give no information about static pools of myosin heads that are not affected in these protocols. This point is now made explicitly in the second paragraph of the Results.

Other issues:

Fig 1c: normalized F. What are the actual force values?

Maximal calcium activated forces for native, NR1C and CR1C exchanged trabeculae were 32.4 mN/mm², 28.9 mN/mm² and 37.4 mN/mm², respectively, which are not significantly different at the $P > 0.05$ level (one-way ANOVA followed by Tukey's post-hoc test). Please see also Supplementary Table S1.

Fig 5: My understanding about the spacing between thick and thin filaments in human cardiac fibers is that there is a 20-nm gap between the heads on the thick filament while the heads are only ~15 nm long. Thus, S2 must leave the thick filament backbone to allow the S1 head to bind to actin in human fibers. If their drawing is the correct spacing in mouse fibers, their model for mice may be consistent, but their model would not fit the human fiber spacings. The authors should speak to this point.

If this is a difference between mouse and human, what would they expect to see if they repeated these experiments with human trabeculae, which would not allow for the Fig 5 interpretation?

We are unaware of published lattice spacing measurements in intact human cardiac muscle. In contrast, lattice spacing measurements in mouse hearts in vivo (37-39 nm; Toh et al., 2006, Biophys J) and isolated intact rat ventricular trabeculae (~37 nm; Irving et al, 2000, AJP Heart Circ Physiol; Brunello et al., 2020, PNAS) are quantitatively consistent with the model presented in Figure 5. Inter-filament spacing is larger, by about 10%, in demembranated rat trabeculae in relaxing conditions, but experiments in which the lattice is compressed by the addition of Dextran (Park-Holohan et al., 2021, PNAS) show that $\langle P_2 \rangle$ for the NR1C probe is almost independent of lattice spacing in this range in relaxing conditions, likely because the folded OFF conformations are intrinsic to the thick filament.

Jim Spudich

Reviewer #3 (Remarks to the Author):

The underlying mechanisms of the healthy heart muscle which contracts strongly to eject blood during contraction (systole) and relax between beats (diastole) while refilling with blood are unknown. Contraction during systole and relaxation during diastole are associated to the shortening and stretching of the heart muscle.

Kampourakis and Irving are leading scientists in this field with a long-standing trajectory. In their manuscript, they propose an structural explanation for the shortening-induced deactivation (SDA) of the heart during diastole which sequentially involves the mediation of the regulatory light chains (RLCs) of the two heads of the myosin molecule, organized in the relaxed state as the blocked and free heads of the myosin interacting-heads motif (IHM).

This proposal is a significant contribution to the literature of this field as they propose a model based on structural information to explain the SDA, focused on the sequential folding of the myosin heads onto the thick filament backbone, being mediated specifically by the regulatory light chains (RLCs) of both myosin heads. This proposal should be of interest to a wider field of readers as it involves the well-known Frank-Starling relationship for the heart, so it should certainly influence the thinking on the field. To support their proposed model, the authors did a comprehensive study of the orientation of bifunctional fluorescent probes, inserted in the RLCs of the IHMs of rat demembrated ventricular trabeculae, during Ca²⁺-induced contraction; together with rapid shortening and re-stretching to mimic SDA.

---It would improve the manuscript if the major points detailed below are addressed.

Major points:

(1) In Fig. 1a the hand of the helix of IHMs shown is reversed. With the IHMs asymmetric conformation as shown in the figure, the sub-fragment 2 (S2) of the myosin tail of the IHM point towards the bare zone at the top, so the helix should run to the right, not to the left.

---This figure (Fig. 1a) should be corrected.

Corrected. Thank you.

Note: As lines are not numbered, the comments below refer to the actual page quoting the specific text.

(2) On the last paragraph of page 3 it is stated that: "We introduced bifunctional sulfo-rhodamine (BSR) probes into either the N- or C-terminal lobe of the RLC, subsequently referred to as NRLC and CRLC respectively, in demembrated trabeculae from rat heart by replacing about half of the native RLC with recombinant cRLCs in which BSR either crosslinked helices B and C in NRLC, or was attached along the E-helix in CRLC (Fig. 1b)".

---The authors should address a possible disruption of the IHM due to the insertion of these probes.

Does this possibility have been ruled out and if so, how? Also, it should be clarified if only one probe or the two probes were inserted on one or both heads of the IHM. As half of the RLCs are replaced, do they know whether these go on randomly, or could they possibly all go onto the free heads, and how would this affect the interpretation?

The first part overlaps with point 2 of reviewer 2 (please see above). Very similar functional parameters for native, and NRLC or CRLC exchanged trabeculae show that RLC exchange had almost no effect on myosin motor function and thick filament-based regulation in demembranated ventricular trabeculae (Kampourakis et al., 2016, PNAS; Kampourakis et al., 2018, JMCC; Zhang et al., 2017, Elife). Moreover, RLC exchange and probe attachment preserves the temperature dependence of the thick filament structure in isolated cardiac muscle previously observed by X-ray diffraction (Park-Holohan et al., 2021, PNAS and Xu et al., 2006, Biophys J). Additionally, probe attachment to RLC had no effect on the ATPase activity of isolated cardiac myofibrils in relaxing conditions (see new Supplementary Figure 1), suggesting the thick filament OFF state is unaffected by the presence of the RLC probes.

We exchanged either NRLC or CRLC probe separately into demembranated trabeculae; the two probes were never present at the same time. This was not explicit in the submitted paper but is now made so at the start of the Results. We showed previously that the value of $\langle P_2 \rangle$ for NRLC is independent of the fraction of RLC exchange in the range from ~10% to >50% using different exchange protocols (Kampourakis et al., 2015, Biophys J). Similar results were obtained for the CRLC probe with an estimated exchange efficiency of >60% (Kampourakis et al., 2018, JMCC). The exchange takes place in rigor conditions in which all myosin heads are attached to actin, so it is likely that the RLC probes become randomly distributed between the free and blocked heads on subsequent relaxation, although we cannot exclude the possibility that the presence of the probe in one head of the myosin dimer might alter the probability that that head became either a blocked rather than a free head. Such a preference would not affect the main conclusions of the paper however, because the NRLC has a similar orientation in the blocked and free heads of the IHM.

(3) For assessing the angles of the probes, the authors measured the fluorescence polarization of the heads in the demembranated muscle by the order parameter . This value should be the average value of the contribution of all the labeled heads present in the measuring beam as shown in the Suppl. Fig. 1a.

a.- In the case of the relaxed state (Fig 1c, pCa low) as it is a demembranated muscle in the relaxed state the IHM should be present, as have been shown either in isolated relaxed thick filaments of mouse and human cardiac muscle (Zoghbi et al. PNAS 2008) (AL-Khayat et al. PNAS 2013) or in demembranated and live relaxed tarantula skeletal muscle (Padron et al PNAS 2020). Therefore, the measured should be the average of the values for each head of the IHM reported by the CRLC and NRLC probes as present in: (1) the blocked head and (2) the free head (when docked on its partner blocked head) completing the IHM. As both heads are roughly parallel to the filament axis then the average for CRLC should be roughly parallel and the one for NRLC roughly perpendicular. So, in the relaxed conditions both probes independently reflect the angles roughly present on the IHM, making the analysis of the probes angles straightforward, with the proviso of discarding the transient changes when the free heads sway away and back (Brito et al. J Mol Biol 2011). However, at higher pCa values when force start to increase and when it reach the plateau, it have been found that in tarantula skeletal muscle during a tetanus the blocked heads remain docked on its S2 near the backbone as it were in the relaxed state, while the free heads are released and bound

to the thin filaments (Padron et al. PNAS 2020). If something similar occurs in the demembrated rat trabeculae when producing force, as the authors propose in their model (Fig. 5), this means that for the higher pCa in Fig. 1c, the measured values could be an average of: (1) the blocked head docked on its S2 reporting a parallel angle from CRLC and a perpendicular one from the NRLC, together with (2) the opposite values from a released free head with a perpendicular angle from CRLC and parallel one from NRLC. So, the measured average value will report mixed contradictory parallel and perpendicular angles from both types of heads.

--The authors should consider and discuss how the previous considerations affect their interpretation of the measured values during force production at high pCa (Fig. 1) as well as on the rapid shortening and re-stretching (Fig. 2, 3, 4).

As explained above in answer to related but less detailed points made by reviewers 1 and 2, and now clarified in Results, these fluorescence polarization measurements report an ensemble average orientation of the probes in terms of the order parameter $\langle P_2 \rangle$. If a fraction of myosin motors stays in the IHM upon calcium activation of cardiac muscle, as indeed seems likely from other experiments referenced above and now discussed further at the end of Discussion, those myosin heads/RLCs will not contribute to the changes in the fluorescence polarization and the measured values of $\langle P_2 \rangle$. Note also that NRLC and CRLC are not present in the same experiment, as clarified in response to the previous point. It follows that the active shortening experiments signal the behaviour of the fraction of myosin heads which are contributing to force generation and active shortening (i.e. actin-attached heads), or myosin heads in the disordered detached state.

For clarification, if all the probes were at the same angle (θ) with respect to the filament axis $\langle P_2 \rangle$ would depend on (θ) as shown below (Dale et al., 1999, Biophys J). However, in practice in any given set of conditions there will be a range of head and probe orientations or multiple populations with different conformations. In that case each population contributes to the overall measured $\langle P_2 \rangle$ in proportion to the fraction of probes in that conformation. An example of this type of analysis (and its limitations) for NRLC and CRLC probes in demembrated trabeculae in near-physiological relaxing conditions is given by Park Holohan et al., 2021.

$$P_2(\cos \theta) = \frac{1}{2}(3 * \cos^2(\theta) - 1)$$

b.- In the three crowns that form the vertebrate cardiac thick filament repeat there are two crowns with three IHMs present on each one, while in the remaining crown the presence of the IHM is

weaker or possibly the six heads it contains are disordered (Zoghbi et al. PNAS 2008) (AL-Khayat et al. PNAS 2013). Does the average coming from probes within the two crowns with IHMs and from the third disordered crown report mixed contradictory parallel and perpendicular angles?

---The authors should consider and discuss how the previous considerations affects their interpretation of the measured .

c.- Accessory proteins like cMyBP-C are present on the C-Zone of the thick filaments but not on the P- and D-zones. Does the average coming from probes within these three zones report mixed contradictory parallel and perpendicular angles?

---The authors should consider and discuss how the previous considerations affects their interpretation of the measured .

The answer to parts b and c is essentially the same as that to a. We agree that the crowns are unlikely to have identical motor conformations in relaxed trabeculae, and that different zones of the filament may also have different motor conformations. But those effects, which have been considered explicitly in recent papers (Brunello et al., 2020, PNAS; Park-Holohan et al., 2021, PNAS), would not affect our interpretation of the changes in $\langle P_2 \rangle$ in response to either addition of calcium or active shortening, they only affect the baseline value of $\langle P_2 \rangle$ from which those transients start.

(4) In the Discussion there are several paragraphs in which knowing how labeled probes are distributed between the blocked and free heads is important as well as how relevant is this distribution on the measured changes, as these changes are averages of reports from probes mixed between both types of heads:

1.- On page 10, first paragraph it is stated that “The CLRC probe, and by inference the orientation of the C-terminal domain of the RLC, tracks the expected fraction of myosin motors attached to thin filaments in all the protocols and on all timescales studied, as expected if CRLC orientation primarily monitors the fractions of attached and detached motors with their distinct conformations.”

--- The authors should consider and discuss how their interpretation of the measured is affected when considering that a fraction of heads could remain bound to the backbone.

*The answer is essentially the same as that to points (2) and (3) above. Our interpretation is only based on the **changes** in probe orientation from a set of starting conformations that represent an ensemble average and cannot be assigned to specific orientation populations using the starting $\langle P_2 \rangle$ values before the perturbation. However, that limitation does not constrain the interpretation as presented in the paper, because that was not based on absolute values of $\langle P_2 \rangle$ but only on (1) their calcium dependence and kinetics and (2) the similarity of NRLC orientation in published cryo-EM structures of the IHM and the actin-bound rigor complex.*

2.- On page 11, second paragraph it is stated that “Flexibility between NRLC and CRLC has also been inferred from cryo-EM structures of both the actin-bound motor in the nucleotide-free or rigor complex⁴¹, and in the folded OFF or IHM state of the myosin dimer^{42,43} (Fig. 1a).”

--- The authors should consider quoting here the papers that addressed the RLC-RLC flexibility which are Brito et al. J. Mol. Biol. 2011 and Alamo et al. J Mol. Biol. 2016.

We have incorporated the requested references into the paper.

3.- On the second paragraph in page 11 it is stated that “Moreover, comparison of the NRLC orientation distribution deduced from the multi-probe studies in trabeculae²⁵ with the cryo-EM structures showed that some NRLCs stay folded against the thick filament backbone in the OFF orientation during active isometric contraction and, unexpectedly, that the OFF orientation of NRLC is similar to that in the actin-bound rigor complex.”, and on the third paragraph “The potential implications of those findings for interpretation of the present results can be illustrated by considering a hypothetical myosin dimer conformation in which one motor is in the actin-attached rigor or post-power stroke state determined by cryo-EM of the nucleotide free actin-motor complex⁴¹ and the other is folded against the thick filament backbone in the ‘blocked’ motor conformation of the IHM determined by cryo-EM of isolated thick filaments^{43,44} (Fig. 5a, upper).” In fact, the presence of the blocked head docked on its S2 and backbone during contraction of tarantula skeletal muscle have been suggested by time-resolved low-angle X-ray diffraction studies (Padron et al. PNAS 2020), as well as in the relaxed state when inducing disordering of the free heads by low temperature (Ma et al J. Gen. Physiol. 2021).

--- The authors should consider quoting the papers that addressed what they stated in the two mentioned paragraphs.

We have incorporated the requested references into the paper.

4- In Fig. 5 (cf. Suppl. Fig. 5) the authors propose a model based on the sequential folding of cardiac myosin heads into the OFF state during active shortening based on the measured average from all heads present in the measuring beam. While the evidence from the probes in the last step (interacting-heads motif) are unambiguous (see point 3a above), the evidence for the first two steps (pre-power stroke and post-power stroke) could have a mixed contradictory origin.

--- The authors should assess the implications that the mentioned mixed contradictory origin has in the first two steps of their model.

The answer is partly related to that to points (2), (3) and (4) above. We do not know the distribution of RLC orientation populations before the perturbation from the $\langle P_2 \rangle$ values, but we have independent evidence about the pre-power stroke conformation from a combination of crystallography and cryoEM; this is where the angles in Fig. 5b come from. We then used those conformations to make the conceptual model in part c. This has now been clarified in the text.

Minor points:

1.- In Fig 2 correct uppercase in (C), it should be (c).

Done. Thank you.

Raul Padron

REVIEWERS' COMMENTS

Reviewer #1 (Remarks to the Author):

The authors have been responsive to my previous comments, and I have no further points to raise.

Reviewer #2 (Remarks to the Author):

The authors have answered all of my questions, and I recommend publication of the revised manuscript. Jim Spudich

Reviewer #3 (Remarks to the Author):

I think the authors has clearly answered and corrected all my points. Thank you.

Note: Concerning the last point (Fig. 5) mentioned by Reviewer #2, I think it would be useful to highlight that the positioning of the IHM on top of the thick filament backbone is only a rough approximation. In the 3D structure of the thick filament solved at higher resolution so far (tarantula, 20 Å) it is clear that the densities of the S2 in the 3D-map emerges from the backbone surface with a slight angle of 6°, causing the helices of IHMs to “float”, separated from the backbone surface by about 2 nm (Fig 3b, Alamo et al J Mol Biol 2016, attached). So in this 3D reconstruction the S2 in the relaxed state is already separated from the backbone surface by about 2 nm.

REVIEWERS' COMMENTS

Reviewer #1 (Remarks to the Author):

The authors have been responsive to my previous comments, and I have no further points to raise.

Thank you.

Reviewer #2 (Remarks to the Author):

The authors have answered all of my questions, and I recommend publication of the revised manuscript. Jim Spudich

Thank you.

Reviewer #3 (Remarks to the Author):

I think the authors has clearly answered and corrected all my points. Thank you.

Note: Concerning the last point (Fig. 5) mentioned by Reviewer #2, I think it would be useful to highlight that the positioning of the IHM on top of the thick filament backbone is only a rough approximation. In the 3D structure of the thick filament solved at higher resolution so far (tarantula, 20 Å) it is clear that the densities of the S2 in the 3D-map emerges from the backbone surface with a slight angle of 6°, causing the helices of IHMs to “float”, separated from the backbone surface by about 2 nm (Fig 3b, Alamo et al J Mol Biol 2016, attached). So in this 3D reconstruction the S2 in the relaxed state is already separated from the backbone surface by about 2 nm.

Done. Thank you.